# URI alleviates tyrosine kinase inhibitors-induced ferroptosis by reprogramming lipid metabolism in p53 wild-type liver cancers

Zhiwen Ding[1,10], Yufei Pan[2,10], Taiyu Shang[3,10], Tianyi Jiang[2], Yunkai Lin[2], Chun Yang[4], Shujie Pang[5], Xiaowen Cui[6], Yixiu Wang[1], Xiao fan Feng ®[2], Mengyou Xu[2], Mengmiao Pei[7], Yibin Chen[2], Xin Li[8], Jin Ding[9], Yexiong Tan[2], Hongyang Wang ®[2,3] ✉, Liwei Dong ®[2] ✉ & Lu Wang[1] ✉

The clinical benefit of tyrosine kinase inhibitors (TKIs)-based systemic therapy for advanced hepatocellular carcinoma (HCC) is limited due to drug resistance. Here, we uncover that lipid metabolism reprogramming mediated by unconventional prefoldin RPB5 interactor (URI) endows HCC with resistance to TKIs-induced ferroptosis. Mechanistically, URI directly interacts with TRIM28 and promotes p53 ubiquitination and degradation in a TRIM28-MDM2 dependent manner. Importantly, p53 binds to the promoter of stearoyl-CoA desaturase 1 (*SCD1)* and represses its transcription. High expression of URI is correlated with high level of SCD1 and their synergetic expression predicts poor prognosis and TKIs resistance in HCC. The combination of SCD1 inhibitor aramchol and deuterated sorafenib derivative donafenib displays promising anti-tumor effects in p53-wild type HCC patient-derived organoids and xenografted tumors. This combination therapy has potential clinical benefits for the patients with advanced HCC who have wild-type p53 and high levels of URI/SCD1.

Worldwide, primary liver cancer is the sixth most commonly diagnosed cancer and the third leading cause of cancer death. Hepatocellular carcinoma (HCC) accounts for 75–80% of all cases of liver cancer[1]. Approximately 50–60% of HCC patients, particularly in advanced HCC, are estimated to receive systemic therapies in their lifespan[2]. Currently, tyrosine kinase inhibitors (TKIs)-based systemic therapies are widely used in HCC patients with Barcelona Clinic Liver Cancer (BCLC)-B/C stage[3], including sorafenib[4], regorafenib[5], lenvatinib[6], and donafenib[7]. Unfortunately, these drugs only moderately improved survival as single agent. Recent studies indicated that combination therapy, such as the combination of TKIs and immune checkpoint inhibitors (ICIs), shows a bright prospect[8]. In the same way, optimal combination therapies and new biomarkers are required to improve the clinical benefits of TKIs-based therapies for HCC.

Sorafenib was the first available TKI for advanced HCC for a decade. However, the objective response rate for sorafenib in SHARP

[1]Department of Hepatic Surgery, Fudan University Shanghai Cancer Center; Department of Oncology, Shanghai Medical College, Fudan University, Shanghai 200032, P. R. China. [2]National Center for Liver Cancer, Naval Medical University, Shanghai 201805, P. R. China. [3]School of Life Sciences, Institute of Metabolism and Integrative Biology, Fudan University, Shanghai 200438, P. R. China. [4]Children's Hospital of Soochow University, Suzhou 215025, P. R. China. [5]Department of Hepatic Surgery V, Eastern Hepatobiliary Surgery Hospital, Naval Medical University, Shanghai 200438, P. R. China. [6]Department of Oncology, Eastern Hepatobiliary Surgery Hospital, Naval Medical University, Shanghai 200438, P. R. China. [7]Department of Hepatic Surgery VI, Eastern Hepatobiliary Surgery Hospital, Naval Medical University, Shanghai 200438, P. R. China. [8]Department of Integrated Chinese and Western Medicine, Eastern Hepatobiliary Surgery Hospital, Naval Medical University, Shanghai 200438, P. R. China. [9]Clinical Cancer Institute, Center for Translational Medicine, Naval Medical University, Shanghai 200438, P. R. China. [10]These authors contributed equally: Zhiwen Ding, Yufei Pan, Taiyu Shang. ✉e-mail: hywangk@vip.sina.com; dlw@smmu.edu.cn; Dr_wanglu01@126.com

study is only 2%[4], suggesting that most HCCs are intrinsically resistant to sorafenib. Recent studies suggest that epigenetics, transport processes, regulated cell death, and the tumor microenvironment are involved in the initiation and development of sorafenib resistance in HCC[9]. Ferroptosis, an iron-dependent form of regulated cell death procedure involving the abnormal metabolism of lipid oxides[10], could be induced by sorafenib in different cancer cells, including HCC. Considering this, the strategy that making cell more vulnerable to ferroptosis might improve the therapeutic efficacy of TKIs[11].

Reprogrammed lipid metabolism of cancer cells not only supplies appropriated energy, but also plays a central role in ferroptosis[12]. Among the metabolic process, the desaturation is of profound interest, since it is supposed to (1) prevent lipotoxicity from excess saturated fatty acyl chains, (2) inhibit ferroptosis triggered by the peroxidation of polyunsaturated fatty acyl chains, and (3) reduce membrane permeability to enhance drug resistance, which are all helpful for cancer cell survival[13]. Stearoyl-CoA desaturase 1 (SCD1) is one of the key enzymes that control the generation of monounsaturated fatty acids (MUFAs). It converts saturated fatty acids (SFAs), especially the palmitoyl-CoA (16:0) or stearoyl-CoA (18:0), to palmitoleic acid (16:1) or oleic acid (18:1), respectively[14]. Therefore, SCD1 activity is critical in maintaining the appropriate ratio of saturated to unsaturated fatty acids within the cell. Previous studies showed that SCD1 is highly expressed in some liver cancers, and its inhibition sensitizes cancer cells to sorafenib[15,16]. However, it remains to be explored how the abnormal expression of SCD1 in HCC is regulated.

Unconventional prefoldin RPB5 interactor (URI) is reported to be involved in the nicotinamide adenine dinucleotide (NAD[+]) metabolism[17], nonalcoholic fatty liver disease[18], and glucose metabolism[19]. Our group has also reported that URI promotes tumorigenesis and chemoresistance of intrahepatic cholangiocarcinoma (iCCA) by modulating oxidative stress[20].

Here, we show the role of URI in the reprogramming of lipid metabolism in HCC, partially by maintaining aberrant SCD1 expression in a p53-dependent manner, which in turn promotes HCC resistant to TKIs-induced ferroptosis. We also provide a strong rationale for testing the combination therapy with SCD1 inhibitors and TKIs for the advanced HCC with wild-type p53.

## Results

### URI promotes resistance to TKIs-induced ferroptosis in liver cancer cells

Firstly, we performed RNA sequencing (RNA-seq) analysis to characterize the URI-dependent transcriptomic alteration. We found that a total of 1676 genes were differentially expressed between shRNA (Ctrl) and shURI HepG2 cells (Supplementary Fig. 1a and Supplementary Data 1). Functional enrichment analyses using the Kyoto Encyclopedia of Genes and Genomes (KEGG) pathway revealed that ferroptosis was one of the top altered pathways regulated by URI (Fig. 1a). The expression of ferroptosis-related genes[10] were increased with URI knockdown (Supplementary Fig. 1b, c). Gene set enrichment analysis (GSEA) revealed that ferroptosis, but not apoptosis signaling, was positively enriched in shURI cells compared with their control. (Fig. 1b and Supplementary Fig. 1d). These results suggest that URI might play an important role in ferroptosis. Meanwhile, pathways, including EGFR tyrosine kinase inhibitor resistance, regulation of lipolysis, p53 signaling pathway, and transcriptional misregulation in cancer, were also regulated by URI (Fig. 1a).

There have been many clinical trials evaluating TKIs for advanced HCC (Table 1)[4–7,21,22]. Consistent with the cytotoxic effect, these TKIs effectively reduced tumor viability among various cancer cell lines tested (Fig. 1c and Supplementary Fig. 2a). Notably, sorafenib, regorafenib and donafenib, with B-RAF as their common targets, favored ferroptosis-like cell death, while other drugs tested showed little effect on cellular lipid peroxidation (Fig. 1d and Supplementary Fig. 2b).

We then tested whether URI regulates TKIs-induced cytotoxicity. JHH1 and HepG2 cells were treated with increasing concentrations of sorafenib for 48 h and cell proliferation was assessed. URI depletion significantly increased the sensitivity of JHH1 and HepG2 cells to sorafenib (Supplementary Fig. 2c), with a decreased IC50 (causing 50% inhibition of viability) from 8.324 μM (Ctrl) to 6.480 μM (shURI) in JHH1 cells, and from 8.047 μM (Ctrl) to 4.069 μM (shURI) in HepG2 cells, respectively (Fig. 1e). Similar patterns of dose-effect curves were also found in response to regorafenib or donafenib between HepG2-shURI and control cells (Supplementary Fig. 2d). Clonal formation assays confirmed the role of URI in sorafenib resistance (Fig. 1f, g). Interestingly, URI knockdown in JHH1 and HepG2 cells substantially increased sorafenib-induced lipid peroxidation (Fig. 1h), suggesting that URI might inhibit the TKIs-induced ferroptosis. Furthermore, sorafenib-induced cytotoxic effect in URI knockdown cells could be rescued when they were pretreated with the antioxidant N-acetylcysteine (NAC) or the ferroptosis inhibitor ferrostatin-1 (Fer-1; lipid peroxidation scavenger), while the inhibitors of apoptosis, necrosis, or autophagy had little effect (Fig. 1i). In addition, URI depletion enhanced RSL3 (a GPX4 inhibitor)-induced ferroptosis as well (Supplementary Fig. 2e). Together, these results suggest that URI is associated with resistance to TKIs-induced ferroptosis in liver cancer cells.

### Ectopic expression of URI reprograms SCD1-associated lipid metabolism in cancer cells

Cancer cells require higher levels of lipid metabolism than normal cells, which can determine the sensitivity of cells to ferroptosis[12]. To examine URI-mediated lipid metabolism and lipidomic change, mass spectrometry-based lipidomic analysis were performed in HepG2-shURI and HepG2-Ctrl cells (Fig. 2a, Supplementary Fig. 3a–c). Our lipidomic data revealed 961 different lipid species, consisting of 219 phosphatidylcholines (PCs), 215 triglycerides (TGs), 106 phosphatidylethanolamines (PEs), and other lipid classes (Supplementary Fig. 3a). The relative content of TGs, the most abundant lipid class mainly containing monounsaturated fatty acid chain, was significantly decreased in URI depleted HepG2 cells than its control, with the decreases of phosphatidylinositol (PI(18:1/18:1)) and diglyceride (DG(16:1/18:1/0:0)) (Fig. 2a, b). In contrast, the relative content of phosphatidylglycerol (PG(16:0/16:0)) was increased in HepG2-shURI cells (Fig. 2b). Meanwhile, PG(16:0/16:0) presented a negative correlation with TG(16:1/16:1/16:1), TG(16:1/18:1/16:1), and PI(18:1/18:1) (Supplementary Fig. 3c). These results suggested that URI depletion changes lipid compositions in liver cancer cells.

We analyzed the fatty acids contents in HepG2 cells. The intracellular lipid peroxidation, a major event in ferroptosis, was determined by the ratio of MUFAs and polyunsaturated fatty acids (PUFAs)[23,24]. We found that the SFAs levels were increased in HepG2-shURI cells than its controls, and the SFAs-derived metabolites MUFAs levels, especially the 16:1(n-7) and 18:1(n-9) MUFAs, were significantly decreased (Fig. 2c). These results strongly suggest that URI knockdown leads to an impaired convention from SFAs to MUFAs. SCD1 is responsible for the synthesis of 16:1(n-7) and 18:1(n-9) MUFAs from the 16:0 and 18:0 SFAs, respectively (Supplementary Fig. 3d). Notably, the16:1(n-7)/16:0 and 18:1(n-9)/18:0 ratios, which are surrogates of SCD1 activity, were significantly decreased in HepG2-shURI cells than their controls (Fig. 2d), indicating a potent association between URI expression and SCD1 activity/expression.

PL-PUFAs are susceptible to ROS and their lipid peroxidation can fuel ferroptosis cascade. On the contrary, MUFAs could suppress this process by promoting the displacement of PUFAs from plasma membrane phospholipids[24]. We then analyzed the lipid species of phospholipids (such as PC, PE, PI) between HepG2-shURI and HepG2-Ctrl cells. There was a decreasing tendency of MUFA in phospholipids of HepG2-shURI cells than controls under steady-state (Fig. 2e). The contents of C16:0/C20:4 PL-PUFA were increased in HepG2-shURI than

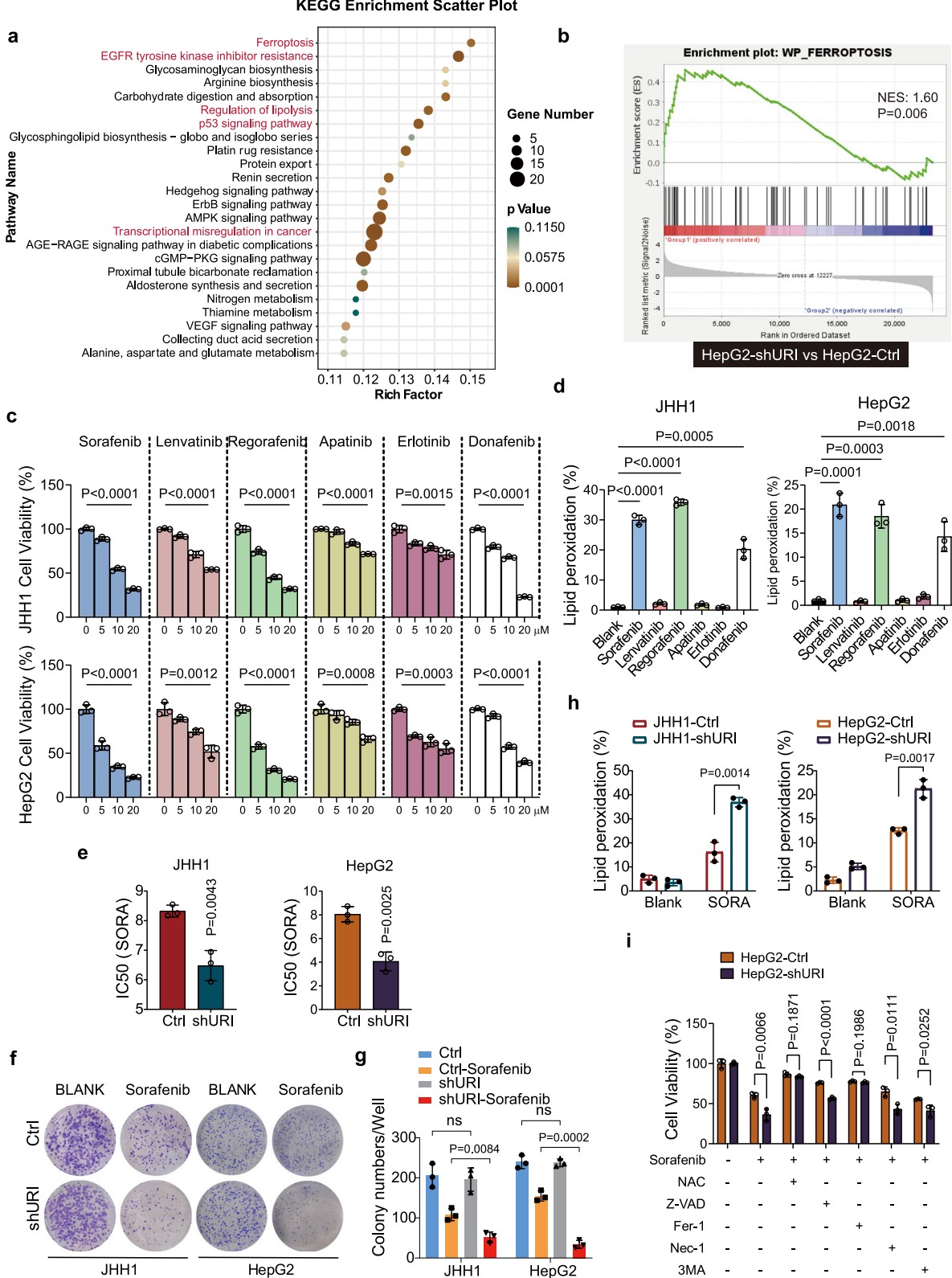

control cells, while the levels of PL-MUFA C16:0/C18:1 were decreased in HepG2-shURI cells (Fig. 2f). Thus, although no significant change in PUFAs was found between HepG2-shURI and control cells, the PL-PUFA was decreased.

De novo fatty acid synthesis (FAS) involves the coordinated actions of several enzymes (Fig. 2g)[25]. We next measured mRNA levels of the key enzymes in FAS pathway in JHH1 and HepG2 cells together

with their URI depletion ones (Fig. 2h and Supplementary Fig. 3e). The transcripts of enzymes participating in saturated fatty acid synthesis or fatty acid elongation, including *SCD1*, *FASN*, *FADS2*, *ACACA*, and *ELOVL6*, were significant reduced in JHH1-shURI and HepG2-shURI cells, comparing with their parental cells (Fig. 2h). Altogether, these results indicate that URI depletion reprograms lipid metabolism in liver cancer cells.

**Fig. 1 | URI depletion promotes TKIs-induced ferroptosis in cancer cells. a** Kyoto Encyclopedia of Genes and Genomes (KEGG) analysis of the differential expressed genes between HepG2-shURI and control cells. The top 24 enriched pathways were listed. The bubble size indicates changed gene numbers and colors represent false discovery rate (*P*-value). **b** Analysis of "WP_Ferroptosis" geneset between HepG2-shURI and control cells by GSEA software (v4.1.0), the NES and FDR *P*-value were shown. **c** Relative viability of JHH1 and HepG2 cells treated with different concentrations of sorafenib, lenvatinib, regarofenib, apatinib, erlotinib or donafenib for 48 h and cell viability was assayed by measuring cellular ATP levels (*n* = 3 biological replicates). **d** HepG2 cells were treated as indicated for 48 h and cells were then stained with 20 μM C11-BODIPY followed by flow cytometry (*n* = 3 biological replicates). Sorafenib, (10 μM); lenvatinib, (10 μM); regarofenib (10 μM) for JHH1 cells and (5 μM) for HepG2 cells; apatinib, (20 μM); erlotnib, (20 μM);

donafenib, (10 μM). **e** IC50 values were calculated according to experiments in Supplementary Fig. 2c. **f, g,** Long-term colony-formation assay of cells treated with or without 2.5 μM sorafenib for 14 days (**f**), and the quantification of clones were shown in (**g**) (*n* = 3 biological replicates). **h** Cells were treated with or without sorafenib (10 μM) for 24 h and lipid peroxidation was measured (*n* = 3 biological replicates). **i** Cells were treated with or without sorafenib (10 μM), antioxidant N-acetylcysteine (NAC, 1000 μM), apoptosis inhibitor Z-VAD-FMK (50 μM), ferroptosis inhibitor ferrostatin-1 (Fer-1, 10 μM), necrosis inhibitor necrostatin-1 (Nec-, 50 μM) or autophagy inhibitor 3-methyladenine (3-MA, 10 μM) as indicated for 48 h, then cell viability was measured (*n* = 3 biological replicates). Data are means ± SEM. Statistical significance in (**c**–**e**) and (**g**–**i**) is determined by two-tailed unpaired *t*-test. Source data are provided as a Source Data file.

## URI promotes resistance to TKIs-induced ferroptosis via SCD1

Lipid metabolic reprograming is involved in cancer drug resistance[26]. Gene ontology (GO) analysis also showed that the lipid catabolic process is enriched in sorafenib-treated HepG2 cells versus its control cells (Supplementary Fig. 3f). To explore whether the protein levels of enzymes involved in lipid metabolism would be changed by URI depletion, we analyzed their levels in various cell lines at basal status. Although significant messenger RNA transcription changes of *ACACA*, *FASN* and *FADS2* in URI-knockdown cells were observed (Fig. 2h), their protein levels were not affected by URI (Fig. 3a). Remarkably, we found lower SCD1 protein levels in shURI cells than their controls (Fig. 3a), which was consistent with the reduction of *SCD1* mRNA levels (Fig. 2h), suggesting that URI could regulate SCD1 activity by affecting its transcription.

By re-analysis the public transcriptional GEO datasets (accession code GSE96793, GSE96794[27] and GSE121153[28]) of sorafenib-treated HCC tumor cells and sorafenib-resistant xenografts, an overlap of 13 genes, including *SCD1*, was identified as the core gene set involved in sorafenib-resistant (Supplementary Fig. 4a). Western-blot assay confirmed the reduction of SCD1 in sorafenib-treated JHH1 and HepG2 cells (Supplementary Fig. 4b). These results suggest a potent role of SCD1 in sorafenib resistance. Interestingly, the ferroptosis inducer, RSL3, also slightly reduced SCD1 expression in tumor cells tested (Supplementary Fig. 4b). The cystine-import-GSH-GPX4 machinery is the canonical pathway in regulating ferroptosis[29]. However, URI depletion induced mild changes of the protein levels of SLC7A11, GCLC, GCLM and GPX4 in JHH1 and HepG2 cells treated with or without RSL3 (Fig. 3a and Supplementary Fig. 4b, c). We found that treatment with RSL3 or sorafenib at a lethal dosage can significantly decrease the protein levels of SCD1 but not GPX4 in wild-type and shURI cells (Supplementary Fig. 4b, c). Thus, we speculated that URI might promote resistance to TKIs-induced ferroptosis by upregulating SCD1 expression. The combination of RSL3 and A939572 or MK8245, both are SCD1 inhibitors, showed synergistic effect in liver cancer cells (Supplementary Fig. 4d). SCD1 inhibitor alone did not induce ferroptosis in Ctrl cells compared with shURI cells (Supplementary Fig. 4e, f). We further explored whether SCD1 inhibitors improved the therapeutic effects of sorafenib on liver cancer cells as well. As expected, the combination of sorafenib and A939572 or MK8245 showed

dramatic cytotoxic effect (Supplementary Fig. 4g, h). Moreover, ferrostatin-1 could reduce cell death induced by sorafenib or the combination therapy both in shURI and control cells (Fig. 3b). Consistent with these results, SCD1 inhibitors in combination with sorafenib showed a synergistic inhibition of tumor cell proliferation in long-term clonogenic assays. Inhibition of ferroptosis could relieve proliferation inhibition induced by this combination treatment (Fig. 3c, d). Further experiments showed that the combination of sorafenib and SCD1 inhibitors significantly elevated lipid peroxidation in cancer cells, the addition of ferrostatin-1 likewise reduced lipid peroxidation induced by the combination therapy (Fig. 3e and Supplementary Fig. 4i). To confirm these findings, we stably transfected MOCK or Flag-SCD1 plasmids into cancer cells. Our results showed that stable expression of Flag-SCD1 in URI knockdown cells was sufficient to re-resist the cells to ferroptosis (Fig. 3f). Meanwhile, supplementation of exogenous SCD1 declined sorafenib-induced lipid peroxidation (Fig. 3g). Altogether, our results reveal that URI promotes resistance to TKIs-induced ferroptosis by upregulating SCD1.

## URI positively regulates SCD1 transcription via wild-type p53 in liver cancer cells

SCD1 has been shown to be regulated both at the transcriptional level by several transcription factors (TFs) and at the post-translational level through ubiquitination and subsequent degradation by the proteasome[14]. Transfecting with His-URI plasmids into shURI cells could rescue SCD1 protein levels (Fig. 4a). To investigate how URI upregulates SCD1 expression in cancer cells, we first examined the effect of URI on the protein stability of SCD1. MG132, a proteasome inhibitor, did not increase the protein levels of SCD1 in shURI cells (Supplementary Fig. 5a). Consistently, the ubiquitination levels of neither endogenous nor exogenous SCD1 proteins were affected by URI expression (Supplementary Fig. 5b, c). These results suggest that URI upregulates SCD1 expression independent of the ubiquitin-proteasome system. Thus, we hypothesized that URI might promote SCD1 expression by upregulating its transcription. Indeed, reporter assay showed that the values of SCD1-luc in shURI cells were significantly lower than that in Ctrl cells (Fig. 4b).

To further identify which transcription factor mediates the upregulation of SCD1 by URI, we transfected siRNA targeting SREBP, ChREBP or LXRα[14,30], the reported main TFs in regulating SCD1, into cancer cells (Supplementary Fig. 5d). The URI-mediated differences of SCD1-luc intensities were not abrogated after interfering these TFs (Fig. 4c). As ChREBP and LXRα were shown to increase SCD1 transcription in a SREBP-dependent manner[31,32], we performed chromatin immunoprecipitation (ChIP) analysis with SREBP antibody in HepG2-Ctrl and HepG2-shURI cells and revealed that multiple binding sites of SREBP in the *SCD1* promoter regions (Fig. 4d, Supplementary Fig. 5e, f and Supplementary Data 2). However, the binding of SREBP with *SCD1* promoter was not affected by URI expression, consistent with the luciferase reporter assay. Our data suggest that these transcription factors are not involved in URI regulating SCD1.

### Table 1 | TKIs for the treatment of advanced HCC

| Kinase inhibitors | Targets | References |
|---|---|---|
| Sorafenib | RAF1, B-RAF, VEGFR1-3, and PDGFRβ | SHARP[4] |
| Lenvatinib | VEGFR1-3, FGFR1-4,PDGFRα, RET, and KIT | REFLECT[6] |
| Regorafenib | VEGFR1-3, PDGFRβ, FGFR1-2, RET, KIT, and B-RAF | RESORCE[5] |
| Apatinib | VEGFR2 | AHELP[21] |
| Erlotinib | EGFR | SEARCH[22] |
| Donafenib | RAF1, B-RAF, VEGFR1-3, and PDGFRβ | ZGDH3[7] |

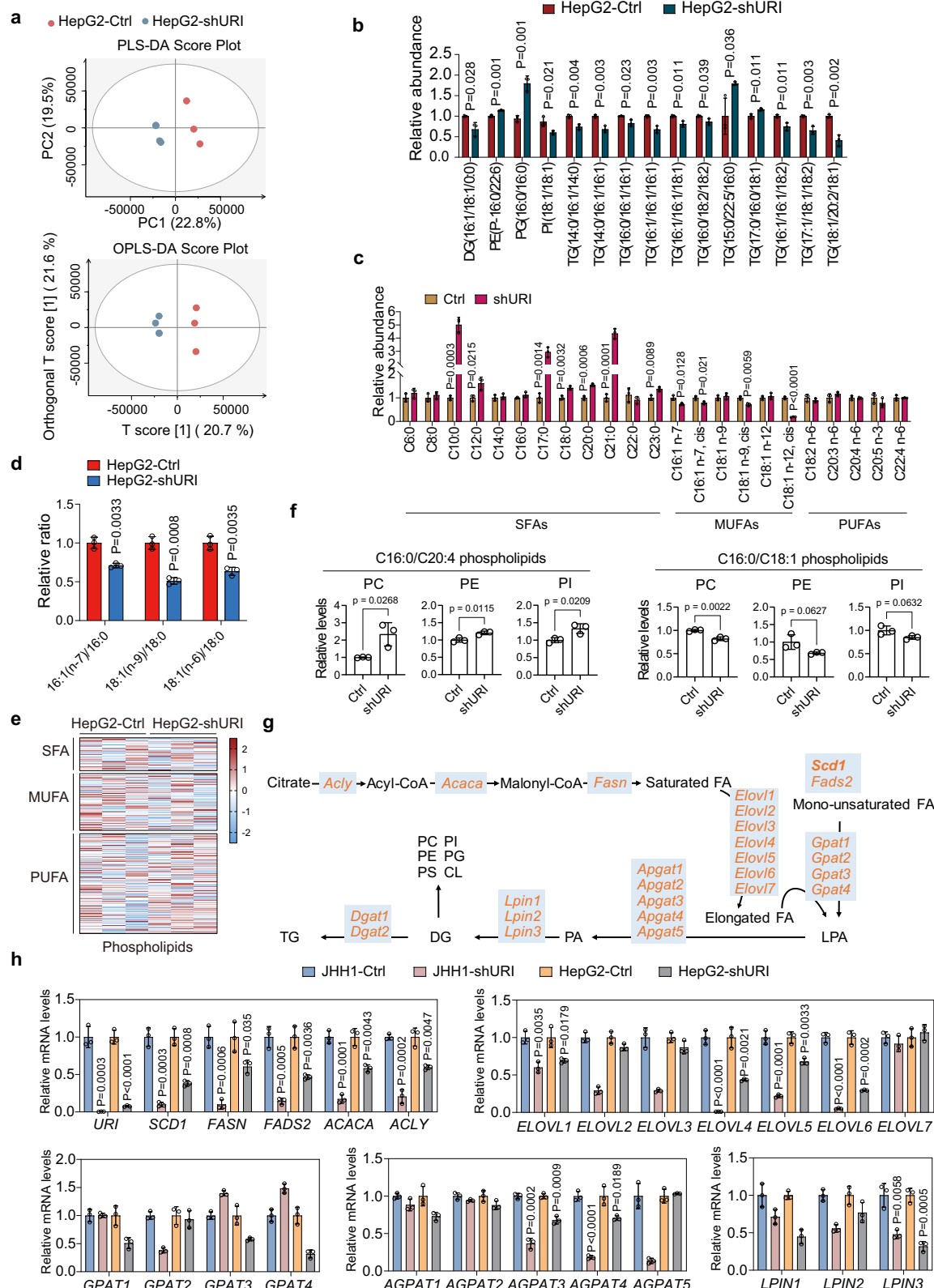

**Fig. 2 | URI depletion altered lipid metabolism of cancer cells. a** Lipid profiles of the cells were compared using partial least squares discriminant analysis (PLS-DA) or orthogonal (O)PLS-DA models. **b** Relative abundance of different diacylglycerol, phospholipids and triacylglycerol that contain C16 and C18 acyl chain are shown on the basis of Supplementary Fig. 3b (*n* = 3 biological replicates). **c** Relative abundance of fatty acid species in HepG2-shURI and control cells (*n* = 3 biological replicates). **d** The indicated MUFA/SFA ratios in HepG2-shURI and control cells (*n* = 3 biological replicates). **e** Heatmap summarized the phospholipids levels between HepG2-shURI and control cells, and the data were shown as z-score. **f** Relative levels of C16:0/C20:4 phospholipids and C16:0/C18:1 phospholipids in HepG2-shURI and control cells (*n* = 3 biological replicates). **g** Schematic of the hepatic de novo lipogenesis pathway, where genes are represented in yellow and metabolites in black. **h** Relative mRNA expression for genes encoding major enzymes in hepatic lipogenesis in the indicated cells (*n* = 3 biological replicates). Data are means ± SEM. Statistical significance in (**b**–**d**), (**f**) and (**h**) is determined by two-tailed unpaired *t*-test. Source data are provided as a Source Data file.

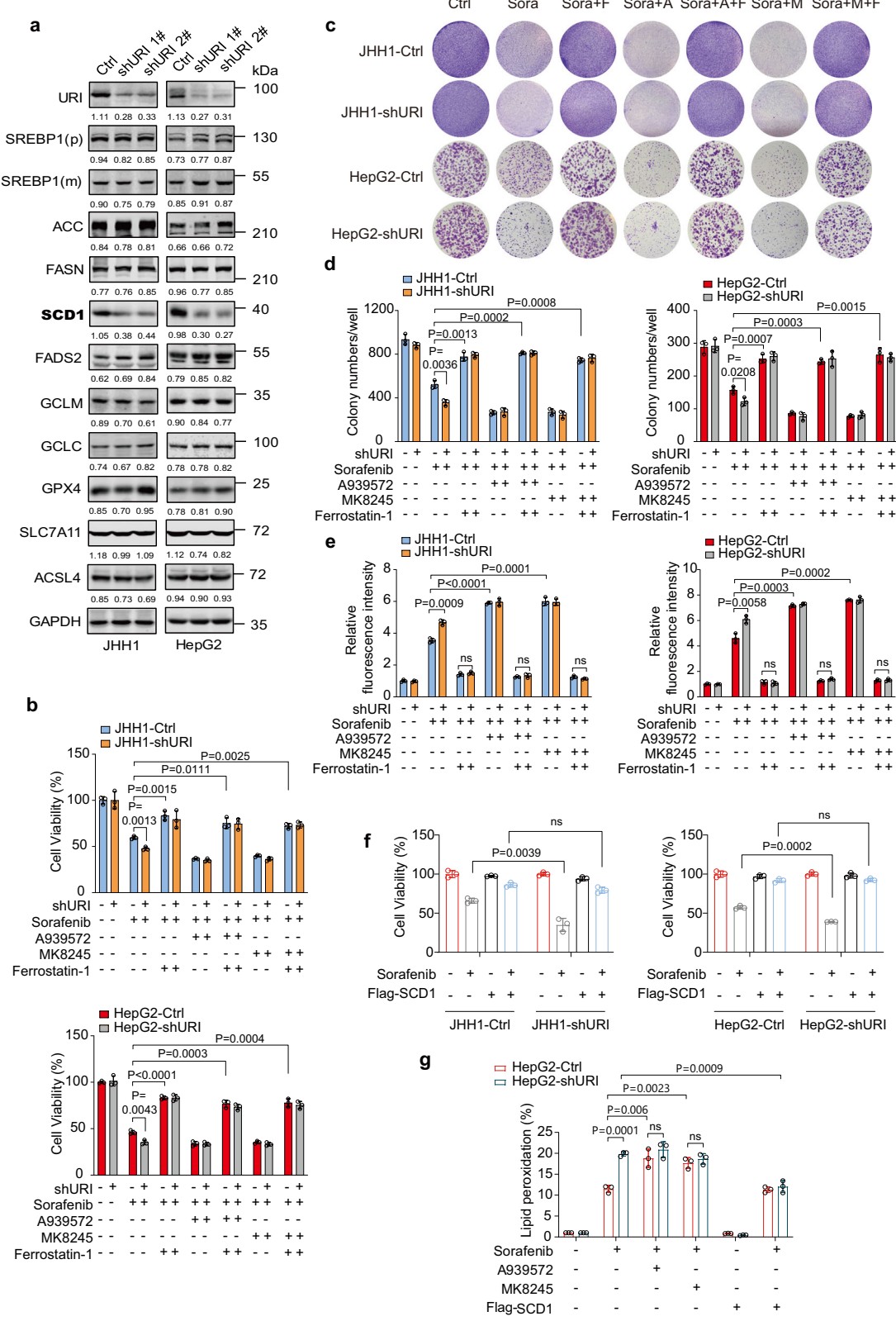

Previous study has suggested that *SCD1* gene is a target of p53-mediated transcriptional repression[33]. Consistently, our CUT&Tag analysis showed the existence of the binding sites for p53 at the promotor region of *SCD1* (Fig. 4e and Supplementary Fig. 5g). GSEA revealed a significant positive enrichment of p53 transcription gene network upon URI depletion (Fig. 4f), which was further confirmed by RT-PCR (Fig. 4g). To validate the role of p53-mediated effects on SCD1

expression, we treated JHH1 and HepG2 cells (which express wild-type p53) with doxorubicin (Dox), a well-known p53-activating agent. The results showed that SCD1 protein and *SCD1* mRNA expression were decreased after Dox treatment (Fig. 4h, i). We next transfected Myc-p53, mutant p53-R273H, or p53-R249S into p53-null cells (Hep3B), and found that wild-type p53 dramatically downregulated the protein levels of SCD1. In contrast, neither DNA-contact mutants (p53-R273H)

**Fig. 3 | URI alleviated TKI-induced ferroptosis in a SCD1-dependent manner.**
**a** Western blotting (WB) of certain lipid metabolism associated enzymes in HepG2 cells infected with lentivirus containing control or two URI-shRNA sequences, respectively. The relative intensities of WB bands were normalized to GAPDH. Three experiments were performed and the representative images from one experiment were shown. **b** Relative viability of cells treated with 10 μM sorafenib in the presence and absence of A939572 (10μM), MK8245 (10 μM), or Ferrostatin-1 (20 μM) for 48 h (*n* = 3 biological replicates). **c**, **d** Long-term colony-formation assay of cells treated with 2.5 μM sorafenib (Sora) in the presence and absence of A939572 (A, 2.5 μM), MK8245 (M, 2.5 μM), or Ferrostatin-1 (F, 5 μM) for 14 days (**c**), and the quantification of three independent assays was shown in (**d**). **e** Liperfluo

assays of JHH1 and HepG2 cells with or without URI knockdown treated with 10 μM sorafenib in the presence and absence of A939572 (10 μM), MK8245 (10 μM), or Ferrostatin-1 (20μM) for 48 h (*n* = 3 biological replicates). **f** JHH1-Ctrl, JHH1-shURI, HepG2-Ctrl, and HepG2-shURI cells were transfected with MOCK or Flag-SCD1 plasmids for 48 h and then treated with 10 μM sorafenib, the relative viability of cells was measured (*n* = 3 biological replicates). **g** HepG2-Ctrl and HepG2-shURI cells transfected with MOCK or Flag-SCD1 plasmids were treated with 10μM sorafenib in the presence and absence of A939572 (10 μM) or MK8245 (10 μM) for 48 h and the lipid peroxidation was measured (*n* = 3 biological replicates). Data are means ± SEM. Statistical significance in (**b**) and (**d–g**) is determined by two-tailed unpaired *t*-test. Source data are provided as a Source Data file.

nor conformational mutants (p53-R249S)[34] affected SCD1 protein levels (Fig. 4j). We explored the potential p53 binding locations and sequences on the human *SCD1* gene (Supplementary Fig. 5h). Indeed, the 5'flanking region of the *SCD1* gene contains one site that matches the consensus p53-binding sequence (Fig. 4k). ChIP analysis further demonstrated that endogenous p53 binds on the promoter region of the *SCD1* gene (Fig. 4l). Moreover, Dox and nutlin-3, both can activate p53, could not reverse the increased protein levels of SCD1 upon p53-knockdown condition (Fig. 4m and Supplementary Fig. 5i). We speculated that URI upregulated SCD1 expression was mediated by p53. URI depletion enhanced p53 binding on the promoter region of the *SCD1* (Fig. 4n). We next transfected siRNA of p53 into HepG2-Ctrl and HepG2-shURI cells and found that knockdown of p53 abrogated shURI-mediated SCD1 repression (Fig. 4n). Our data showed that URI depletion significantly decreased SCD1 expression in wild-type p53 cell (JHH1 and HepG2) (Fig. 3a and Fig. 4p). Similar results were also observed in other human cancer cells expression wild-type p53 (HCT116), but not in p53-null (Hep3B) or p53 mutant cells (HUH7, PLC/PRF/5 and HT29) (Supplementary Fig. 5j). In p53 mutant cancer cells, URI depletion also did not affect p53 targeted genes (Supplementary Fig. 5k). Together, these data suggest that URI promotes transcriptional levels of SCD1 mediated by repressing wild-type p53.

## URI promotes ubiquitination and degradation of wild-type p53 in cancer cells

We next transfected His-URI plasmids into JHH1 and HepG2 cells and found that this treatment led to significant reduction of p53 proteins, even in the presence of Dox (Fig. 5a). To explore the molecular mechanisms of the decreased expression of p53 mediated by URI, we first examined the mRNA levels of p53 in Ctrl and shURI cells. Our data showed that URI depletion could not affect the transcriptional levels of p53 (Fig. 5b). Comparable mRNA levels of SREBP and ChREBP were also found (Supplementary Fig. 6a). Stress-induced p53 activation largely occurs through protein stabilization[35]. We further explored whether URI regulates p53 protein stability. Indeed, the half-life of p53 was prolonged in JHH1-shURI cells compared with their control cells by using cycloheximide (CHX) (Fig. 5c). URI depletion significantly decreased ubiquitination of wild-type p53 in HepG2 cells (Fig. 5d).

Since URI is not an E3 ubiquitin ligase, we hypothesized that URI might recruit the E3 ubiquitin ligase to mediate ubiquitination of p53. We further performed liquid chromatography tandem-mass spectrometry (LC-MS/MS) analysis and identified potential URI binding proteins, including TRIM28, USP5, USP7 and USP14, which were reported to regulate ubiquitination of p53[36–39] (Fig. 5e and Supplementary Data 3). To validate the results of the LC-MS/MS, candidates were tested for binding with URI by an immunoprecipitation assay. The results showed an interaction between URI and TRIM28 (Fig. 5f), consistent with previous studies[40]. Two different siRNAs targeting TRIM28 increased p53 protein levels particularly in shURI cells (Fig. 5g and Supplementary Fig. 6b). Furthermore, knockdown of TRIM28 resulted in decreased ubiquitination of p53 with MG132 treatment (Fig. 5h and Supplementary Fig. 6c). As TRIM28 binding with MDM2 contributes to p53 protein stability, we explored whether URI could

affect the interaction between TRIM28 and MDM2. Immunoprecipitation revealed that URI depletion inhibited the formation TRIM28-MDM2 (Fig. 5i and Supplementary Fig. 6d, e). In contrast, over-expression of URI promotes the cooperation of MDM2 and TRIM28 (Fig. 5j). Moreover, URI depletion decreased the complex of TRIM28-MDM2-p53 (Fig. 5i, j and Supplementary Fig. 6d–f), suggesting that URI might recruit E3 ubiquitin ligase TRIM28 and promote the effect of TRIM28-MDM2 on p53 protein ubiquitination. In vitro, GST-pull down assays were performed to confirm whether URI directly interact with TRIM28. The interaction between purified GST-URI and His-TRIM28 increased significantly in a concentration-dependent manner (Fig. 5k). Together, our results suggest that URI interacts with TRIM28 and promotes the formation of TRIM28-MDM2-p53 complex, subsequently ubiquitinating p53. Next, we determined whether modulation of p53 protein levels by URI/TRIM28 had a functional impact on ferroptosis. Knockdown of TRIM28 significantly promotes sorafenib-induced cell death and lipid peroxidation (Fig. 5l–n). In addition, most cancer cells with p53 mutation showed resistant to TKIs in CCLE database[41], suggesting that wild-type p53 might sensitize cells to TKIs and transcriptional repression of SCD1 by p53 might be one of the mechanisms (Supplementary Fig. 6g–i).

## SCD1 inhibitor synergizes with donafenib in liver cancer

Our aforementioned data showed that inhibition of SCD1 significantly reversed the resistance to TKIs of liver cancer cells. We speculated that combination of TKIs and SCD1 inhibitors might be efficient in p53 wild-type cancer. Aramchol, a novel partial inhibitor of SCD1, has been proved as a promising therapy for nonalcoholic steatohepatitis (NASH) and led to no severe adverse events in a phase 2b clinical trial, compared with other SCD1 inhibitors[42]. To evaluate the safety and anti-tumor activity of aramchol in vivo, we utilized a xenograft model of rodents (Supplementary Fig. 7a). Aramchol did not cause weight loss of nude mice (Supplementary Fig. 7b). Moreover, there was no therapeutic effect of aramchol alone on subcutaneous tumors (Supplementary Fig. 7c–e). Indeed, the observed changes of aspartate aminotransferase (AST) indicated a benefit of aramchol in improving liver injury (Supplementary Fig. 7f, g). In addition, the levels of serum TG were not changed on a normal diet (Supplementary Fig. 7h). Aramchol did not cause weight loss or induce significantly pathologic changes in the major organs of mice, including the liver, spleen, kidney, lung and heart (Supplementary Fig. 7i–k). Altogether, these results suggest the safety of aramchol in mice models.

Aramchol had no potent inhibitory function on cell viability in vitro (Fig. 6a), consistent with the results in xenograft models. Remarkably, aramchol enhanced sensitivity of p53 wild-type cancer cells to donafenib, a deuterated sorafenib derivative[7] showed superiority over sorafenib, in long-term clonogenic assays (Fig. 6b, c). We next investigated whether aramchol could show a synergistic effect with donafenib in HCC patientderived organoids (PDOs). The p53 status in our PDOs was also identified (Fig. 6d). Similar with the results in cancer cell lines, URI depletion significantly promoted the sensitivity of donafenib in p53 wild-type PDOs. The combination of donafenib and aramchol indeed exhibited more potent cytotoxic

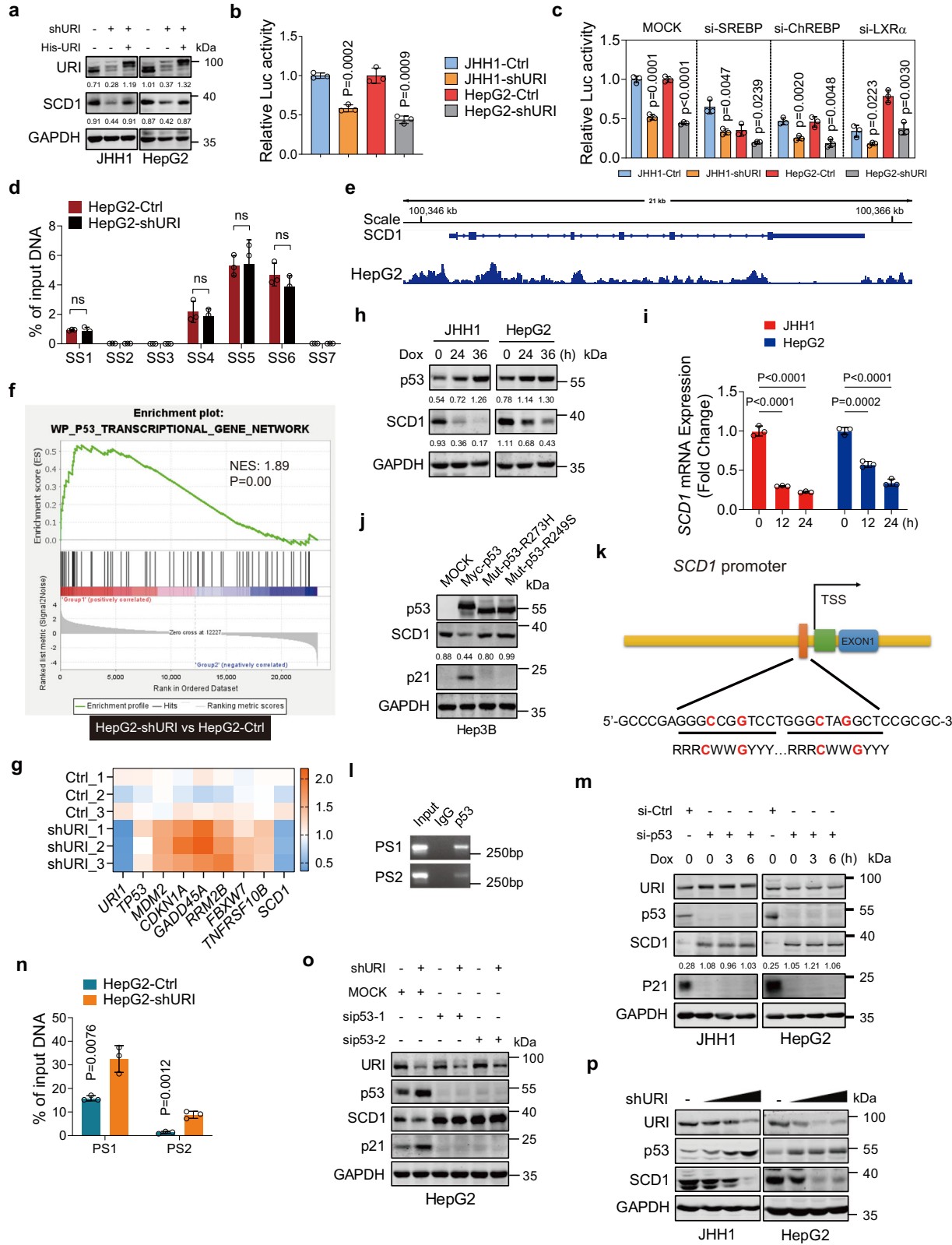

effects than donafenib alone (Supplementary Fig. 8a and Fig. 6e–g). Furthermore, the combination therapy stimulated much higher levels of lipid peroxidation compared to single drugs, suggesting that aramchol promotes donafenib-induced ferroptosis in pre-clinical models (Fig. 6h and Supplementary Fig. 8b). In contrast, p53 mutant PDOs revealed much more resistant to donafenib. The combination showed less effect in p53 mutant PDOs compared with p53 wild-type PDOs

(Supplementary Fig. 8c, d). Although, the combination therapy was able to elevate lipid peroxidation levels (Supplementary Fig. 8e, f).

We further explored the therapeutic effect of the combination treatment in vivo (Fig. 7a). The combination of aramchol and donafenib elicited a complete inhibition of tumor growth in mice xenografted with HepG2-Ctrl and HepG2-shURI cells, whereas the donafenib single treatment played an effective therapeutic role in xenografts with

**Fig. 4 | URI promotes SCD1 transcription by inhibiting p53. a** Representative WB images of cells infected with shURI-lentivirus or transfected with His-URI plasmids. **b** Luciferase assay of SCD1 transcriptional activity in cells under steady state (*n* = 3 biological replicates). **c** Cells were transfected with si-SREBP, si-ChREBP, si-LXRα for 48 h, and the luciferase activity was measured (*n* = 3 biological replicates). **d** ChIP-qPCR was performed on HepG2-Ctrl and HepG2-shURI cells using specific primers flanking the SREBP binding sites (SS) (*n* = 3 biological replicates). **e** The p53 binding sites at the *SCD1* promoter region according to the p53 CUT&Tag data. **f** GSEA analysis showed that "WP_p53 transcriptional gene network" was enriched in the HepG2-shURI cells, the FDR *P*-value and NES were calculated by GSEA software (v4.1.0). **g** Relative mRNA expression of p53 related genes in HepG2-shURI and control cells. **h** Representative WB of cells treated with 10μM Dox at the indicated time. **i** Relative *SCD1* mRNA expression in JHH1 and HepG2 cells treated with 10 μM Dox for the indicated time (*n* = 3 biological replicates). **j** Representative WB of Hep3B cells transfected with wild-type or mutant p53 plasmids. **k** Schematic diagram presenting potential p53 binding location and sequence on human *SCD1* gene. (R, A/G; W, A/T; Y, C/T; nucleotides C and G in red are essential for p53 binding). TSS, transcription start site. **l** ChIP assay in HepG2 cells and qPCR was performed using specific primers flanking the p53 binding sites (PS1, PS2), the representative images of three independent experiments were shown. **m** Representative WB of cells with p53 knockdown treated with 10μM Dox at the indicated time. **n** ChIP-qPCR of the p53 binding sites (PS1, PS2) in HepG2-shURI and control cells (*n* = 3 biological replicates). **o** Representative WB of cells transfected with MOCK or si-p53 for 48 h. **p** Representative WB of cells infected with gradient doses of shURI lentivirus (MOI, 0, 5, 10, 20) for 48 h. The relative intensities of WB bands were normalized to GAPDH in each experiment. WB were performed at least three independently repeated. Data are means ± SEM. *P*-values are determined using two-tailed unpaired *t*-test (**b**–**d**, **i**, **n**). All blots were repeated in three independent experiments and the representative images were shown. Source data are provided as a Source Data file.

HepG2-shURI cells (Fig. 7b–d). In addition, aramchol, alone or in combination with donafenib, has no effect on body weight of mice (Supplementary Fig. 9a). Tumor cell viability was further assessed to evaluate the anti-tumoral effect of the treatments. The combination treatment reduced tumor viability in both of the HepG2-Ctrl and HepG2-shURI models (Supplementary Fig. 9b, c). Immunohistochemical analysis of tumor sections further revealed decreased Ki67 in tumors from mice treated with aramchol and donafenib. Moreover, combined aramchol and donafenib was found to promote synergistic loss of SCD1 and p53 (Fig. 7e, f). The combination also showed an inhibition of tumor growth in mice xenografted with PLC/PRF/5 cells, a p53-R249S mutant human HCC cell line (Supplementary Fig. 9d). However, the tumor suppression effect of the combination treatment was poorer than that in the p53 wild-type models (Supplementary Fig. 9e–g). Combination therapy did not cause weight loss (Supplementary Fig. 9h), indicating good tolerability. Similarly, the combination might promote lipid peroxidation in vivo (Supplementary Fig. 9i). Together, these results showed that the SCD1 inhibitor aramchol is synergetic lethal with donafenib in liver cancer.

Considering that clinically, TKIs is mainly used in patients with advanced HCC. We delayed administration to allow tumors of HepG2 xenografts to grow to a proper size (Fig. 7g). Again, the combination elicited marked tumor control (Fig. 7h, i). In addition, the levels of lipid peroxidation in tumors was significantly elevated with combination treatment (Fig. 7j). Hence, the combination of donafenib and aramchol may achieve significant clinical benefit, particularly in HCC patients with wild-type p53.

## Combined higher expression of URI and SCD1 correlated with worsen prognosis and sorafenib-resistance in HCC

In our previous study[43], we found that URI high-expression in HCC is associated with cancer malignancy and poor survival of patients. Similar results were found in HCC samples from the Gene Expression Profiling Interactive Analysis (GEPIA) database[44] (Supplementary Fig. 10a, b). To further confirm the functional link between the URI-SCD1 axis and clinical cancer therapy, we first utilized a tissue microarray of cohort consisting with 134 advanced HCC samples (cohort A, Supplementary Data 4). We performed multiplex immunohistochemistry/immunofluorescence (mIHC/IF) in these samples to test URI and SCD1 expression, and H-score method was employed to quantitate their expression as described in the material and method (Fig. 8a). URI expression showed a positive correlation with SCD1 expression in cohort A samples (Fig. 8b, c). High expression of URI was associated with high levels of tumor marker α-fetoprotein (AFP) (Supplementary Fig. 10c). Patients with high URI or SCD1 expression were correlated with shorter overall survival (OS) (Supplementary Fig. 10d, e). Furthermore, we stratified patients into four groups according to the expression of URI and SCD1. Significant differences in OS among different subclasses were illustrated by Kaplan–Meier curves (*p* < 0.0001,

log-rank test) (Fig. 8d). Patients with synergetic higher URI and SCD1 expression were at the highest risk with 5-year OS rate of 4.3% (HR, 2.65; 95% CI, 1.76–3.98) compared with 18.8% (HR, 0.38; 95% CI, 0.25–0.57) for lower URI and SCD1. These results indicate that the combination of URI and SCD1 expression is a highly effective predictor for poor prognosis of HCC.

Since the unavailability of p53 status data in our cohort A, to further investigate the role of URI-SCD1 in HCC patients with different p53 status, we employed a new HCC cohort enrolled by Gao et.al, which we named as "Fudan_HCC_cohort"[45]. By analysis of the WES and transcriptome data of this cohort, we found that in HCC patients with p53-WT status, the *SCD1* expression was lower in *URI*^low tumors than in *URI*^high tumors, while other ferroptosis-associated molecules, such as *ACSL4, ALOX12, GPX4, SLC7A11,* and *AIFM2*, were not significantly altered (Fig. 8e). Notably, the *SCD1* level in p53-mutation HCC patients were comparable between *URI*^low and *URI*^high tumors (Supplementary Fig. 10f). Moreover, higher *URI* or *SCD1* expression in p53-WT HCC patients were correlated with poorer clinical outcome (Fig. 8f, g). We did not observe this correlation in p53-mutation HCC patients (Supplementary Fig. 10g, h). Taken together, these results demonstrated that the potential correlation between URI-SCD1 and the clinical outcome of HCC patients exists in patients with wild-type p53, but not in p53-mutation ones.

To further explore whether the role of URI-SCD1 in sorafenib resistance is clinically relevant, the expression of URI and SCD1 was examined in a cohort of HCC patients received adjuvant sorafenib treatment (cohort B) (Supplementary Fig. 11a and Supplementary Data 5). Our data showed that URI expression was closely associated with SCD1 expression, consistent with the results from cohort A (Supplementary Fig. 11b). As most HCC recurrences occurred within 1 year after liver resection, we next analyzed whether URI and SCD1 expression is associated with 1-year recurrence free survival rate (RFS) of patients received adjuvant sorafenib treatment (Supplementary Fig. 11c, d). The 1-year RFS rate of patients with high URI was 20.9% (hazard ratio (HR), 2.21; 95% CI, 1.35–3.63) compared with 53.3% (HR, 0.45; 95% CI, 0.28–0.74) for low URI (*p* = 0.0051, log-rank test). Moreover, the 1-year RFS rate of patients with high SCD1 was 21.7% (HR, 1.81; 95% CI, 1.11–2.95) compared with 48.6% (HR, 0.55; 95% CI, 0.34–0.90) for low SCD1 (*p* = 0.0222, log-rank test). Importantly, patients with combined higher URI and SCD1 expression were at the highest risk with 1-year RFS rate of 20.0% (HR, 3.00; 95% CI, 1.72–5.22) compared with that of 64.0% (HR, 0.33; 95% CI, 0.19–0.58) for low URI and low SCD1 (Supplementary Fig. 11e).

We then employed our previous cohort (named cohort C) which enrolled HCC patients with recurrent HCC, the patients were then received systemic therapy containing sorafenib[46]. The mutation landscape of this cohort was performed (Supplementary Data 6). Forty-five patients were p53-WT and one patient harbored p53 synonymous mutation, whom we also grouped into p53-WT. The protein levels of

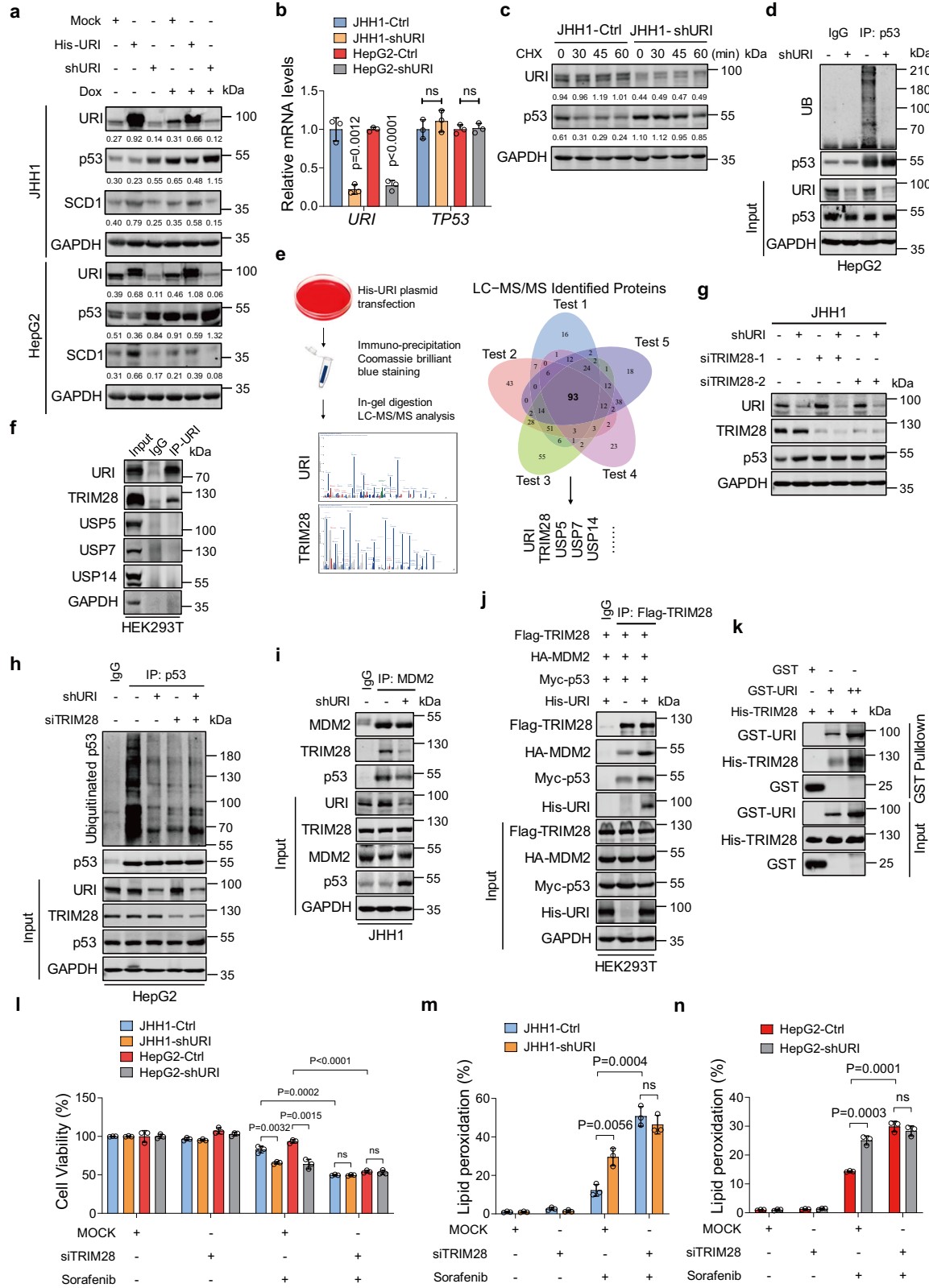

SCD1, URI and p53 were measured by immunohistochemistry. The overall survival of this cohort was higher in patients with p53-WT than p53-mutation ones (Fig. 8h). We found significant correlation between SCD1 and URI in p53-WT group, but not in p53-mutation group (Fig. 8i–k, Supplementary Fig. 11f). Meanwhile, higher levels of SCD1 or URI were associated with worsen prognosis in p53-WT HCC patients receiving sorafenib treatment (Fig. 8i–o), while no significant correlation were found in p53-mutation patients (Supplementary Fig. 11g, h). Thus, our results demonstrated the important role of URI-SCD1 axis in sorafenib resistance in p53-WT HCC patients.

## Discussion

Single-agent TKIs provide modest clinical benefits in unresectable HCC with only 9.2–24.1% objective responders according to

**Fig. 5 | URI promotes p53 ubiquitination and degradation by interacting with TRIM28. a** Representative WB of cells infected with shURI lentivirus or transfected with His-URI plasmid and then treated with 10 μM Dox for 24 h. **b** Relative mRNA expression of *URI* and *TP53* in JHH1 and HepG2 cells with or without URI knockdown (*n* = 3 biological replicates). **c** Representative WB of cells treated with 50 ng/mL CHX at the indicated time. **d** Representative WB of cells transfected with HA-tagged ubiquitin for 48 h and immunoprecipitation with anti-p53 or IgG antibodies. **e** HEK293T cells were transfected with His-URI plasmid before anti-His immuno-precipitation and gel cutting. LC-MS/MS was performed to identify URI potential binding proteins (*n* = 5 biological replicates). **f**, Representative WB of HEK293T cells transfected with His-URI plasmid and immunoprecipitated with anti-His antibody. **g** Representative WB of cells transfected with MOCK or different si-TRIM28 oligos for 48 h. **h** Cells transfected with HA-tagged ubiquitin and with or without siTRIM28 for 48 h and then treated with 10μM MG132 for 6 h, anti-p53 immunoprecipitation

were performed and the indicated proteins were measured. **i** Cell lysates were subjected to anti-MDM2 immunoprecipitation. Immunoblotting was performed with indicated antibodies. **j** HEK293T cells were co-transfected with the indicated plasmids for 48 h, and cell lysates were subjected to anti-Flag immunoprecipitation. **k** GST-pull down shows a dose-dependent direct interacting between His-TRIM28 and GST-URI. **l** Cells were transfected with MOCK or siTRIM28 for 48 h and then treated with 10 μM sorafenib for 24 h, and cell viability was measured (*n* = 3 biological replicates). **m**, **n** Cells were transfected with MOCK or siTRIM28 for 48 h and then treated with 10 μM sorafenib for 24 h, and the lipid peroxidation was measured (*n* = 3 biological replicates). All blots were repeated in three independent experiments and the representative images were shown. The values under the panels indicate the quantification of the bands normalized to GAPDH. Data are means ± SEM. Statistical significance in (**b**) and (**l**–**n**) is determined by two-tailed unpaired *t*-test. Source data are provided as a Source Data file.

mRECIST 1.1[4–7]. To improve the TKIs therapeutic effect, it is urgent to explore the TKIs resistance mechanism and identify therapeutic targets. In this study, by employing various HCC cell lines and patient-derived organoids, we found that URI links TKIs-induced ferroptosis with lipid metabolism reprogramming in a p53-SCD1 dependent manner. URI directly interacts with TRIM28, which leads to ubiquitination and degradation of wild-type p53, preventing the transcriptional inhibition of p53 on *SCD1* and promoting the generation of desaturation products. Subsequently, SCD1-dependent desaturation decreases TKIs-induced ferroptosis and enhances intrinsic resistance of cancer cells to TKIs. Attenuation to the URI-p53-SCD1 axis through URI knockdown or SCD1 inhibitors effectively enhances TKIs-induced ferroptosis in vitro and synergizes the efficacy of TKIs therapy in vivo.

Previous studies have shown that URI is involved in HCC development and various metabolic processes[17–19]. Here, we present an insight into the role of URI in liver cancer cells lipid metabolism reprogramming. URI significantly decreased TKIs-induced ferroptosis without affecting GPX4 pathway, which is a classic ferroptosis regulation pathway. Unrestrained lipid peroxidation is the hallmark of ferroptosis. MUFAs and PUFAs are both participated in the regulation of lipid peroxidation and are key players in determining the sensitivity of cells to ferroptosis. Increased the ratios of MUFA to SFA or PUFA in cancer cell membranes result in reduced lipotoxicity and suppressed susceptibility to ferroptosis[13,29]. Meanwhile, when testing the lipid species in the phospholipids of cells, we found a replacement of PL-MUFA to PL-PUFA. Since MUFAs are mainly produced with desaturation, we propose that desaturation process upregulated by URI, followed by maintaining the fluidity of cancer cell membrane, may be a prerequisite for resistance to ferroptosis.

In humans, SCD1 is the rate-limiting enzyme of desaturation and MUFA synthesis. It introduces a cis double bond at the C9 position of SFAs to MUFAs[14]. Here, we found that URI promotes SCD1 activity, consistent with the elevated MUFAs. The combination of SCD1 inhibitors and TKIs can efficiently abrogate the TKIs resistance mediated by URI. In contrast, exogenous SCD1 enhanced TKIs resistance even in URI knockdown cancer cells. The findings imply that URI promotes TKIs-resistance in a SCD1-dependent manner. Indeed, SCD1 was found to be upregulated in HCC[15,16]. The participation of SCD1 in ferroptosis resistance has also been established[47,48]. Of note, we found that SCD1 expression is positively correlated with URI expression in HCC tissues, especially in p53-WT tumors. Moreover, according to the results of our clinical cohorts, although HCC patients with p53-wild type had a better clinical outcome than p53-mutation group when treated with sorafenib-based therapy, higher levels of URI in the p53-wild type group still indicated worsen prognosis. However, in HCC cells and tissues with p53-mutation, we did not find the effect of URI in sorafenib-resistance. Thus, our results strongly suggest the role of URI in sorafenib resistance is relied on the function of wild type p53. Considering that more than 30% patients in our different cohorts were

URI^high^p53^WT, our results may have good clinical application scenarios, which require further investigation.

The wild-type p53 protein and its cellular pathways mediate tumor suppression, and activated p53 transcriptionally regulates lots of genes which are involved in multiple biological processes[49]. Mutations in p53 gene (21%), which abrogate the tumor suppressor activities of p53 proteins, are the second most common single gene alterations in HCC[50,51]. Approaches towards p53-based therapy are mainly made to develop drugs that target selectively p53 mutants. Moreover, therapeutic strategies that protect p53 from its negative regulators are also attractive[52]. In this study, we identified that URI inhibits the functionality of wild-type p53 in TKIs-induced ferroptosis. It is well known that the E3 ubiquitin ligase MDM2 and p53 proteins form a central hub that receives stressful inputs via MDM2 and respond via p53. TRIM28 interacts with MDM2 and suppresses p53 activation by decreasing the half-life of the p53 protein and rapidly inhibiting it for transcription[36]. We found that URI directly binds with TRIM28, promotes p53 ubiquitination and degradation in a TRIM28-MDM2 dependent manner and subsequently attenuates p53 transcriptional repression targeted SCD1. This may be helpful to keep p53 levels low as has been detected in cancer cells. In addition, HCC with mutant p53 showed poorer clinical responses to TKIs than wild-type p53, which also suggest that strategies to restore p53 levels or improve downstream of p53 pathway might promote the therapeutic sensitivity of cancer cells to TKIs.

Restoring the functionality of p53 in tumors as a therapeutic strategy has mostly met with limited success. Considering the p53 currently undruggable and the subsequent clinical conversion, we focus on the SCD1 activity targeted by p53. The combination of atezolizumab and bevacizumab displayed an acceptable side-effect profile and further improved the objective response rate for systemic treatment of advanced HCC to 33.2%. Therefore, combination therapies are expected to change the landscape of HCC systemic treatment. Unlike other SCD1 inhibitors, aramchol does not result in serious side effects. The efficacy and safety of aramchol has been evaluated in patients with NASH by a randomized, double-blind, placebo-controlled phase2b trial[42]. The phase 3 program is ongoing and open-label part of the study was reported to meet its objectives (NCT04104321). Aramchol is also safe in nude mice, but single-agent has no significant antitumor effect. Among the TKIs that can induce ferroptosis, donafenib is associated with a significantly longer median OS than sorafenib[7]. We found that aramchol combined with donafenib showed a dramatic inhibition of tumor in vivo. Moreover, HE staining of xenograft sections showed that the tumor necrotic area with the combination treatment was significantly enlarged, mimic with the pathological response of clinical treatment. Our results indicate that the combination of donafenib plus aramchol may especially benefit patients who have wild-type p53 HCC tumors.

In summary, URI keeps low levels of p53 in a TRIM28-MDM2 dependent manner, maintains SCD1 activity and accumulation of MUFAs, and subsequently promotes resistance to TKIs in cancer cell.

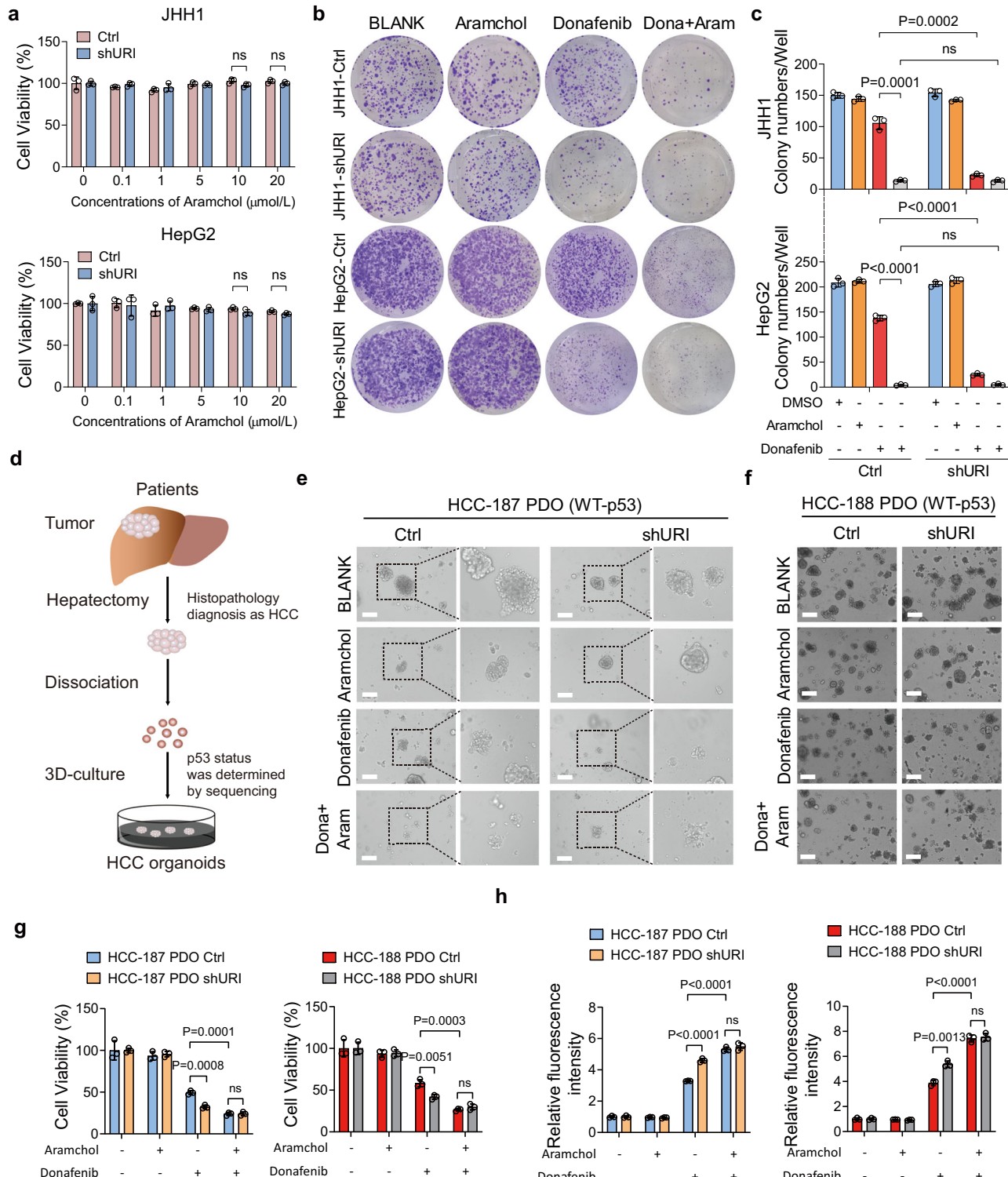

**Fig. 6 | SCD1 inhibitor aramchol enhances the efficacy of donafenib in cancer cells and in patient-derived organoids. a** Cells were treated with aramchol at the indicated concentrations for 48 h, and cell viability was measured ($n = 3$ biological replicates). **b**, **c** Effect of aramchol, donafenib or combination therapy in colony-formation assay of JHH1 and HepG2 cells with or without URI depletion (**b**), and the quantification of three independent assays were shown in (**c**). **d** Schematic representation of HCC patient-derived organoids (PDOs) establishment.
**e**, **f** Representative images of wild-type p53 HCC-187 PDOs (**e**) and wild-type p53 HCC-188 PDOs (**f**) with or without URI depletion treated with aramchol (10 μM), donafenib (10 μM) or combination therapy for 5 days. Scale bar: 200 μm. The experiments were performed with three independent replications, five individual different fields of per well were acquired for every experiment and the representative images were shown. **g** Relative viability of HCC-187 and HCC-188 PDOs with or without URI depletion treated with donafenib in the presence and absence of aramchol for 5 days ($n = 3$ biological replicates). **h** Liperfluo assay of HCC-187 and HCC-188 PODs with or without URI depletion treated with donafenib in the presence and absence of aramchol for 5 days ($n = 3$ biological replicates). Data are means ± SEM. Statistical significance in (**a**), (**c**), (**g**) and (**h**) is determined by two-tailed unpaired *t*-test. Source data are provided as a Source Data file.

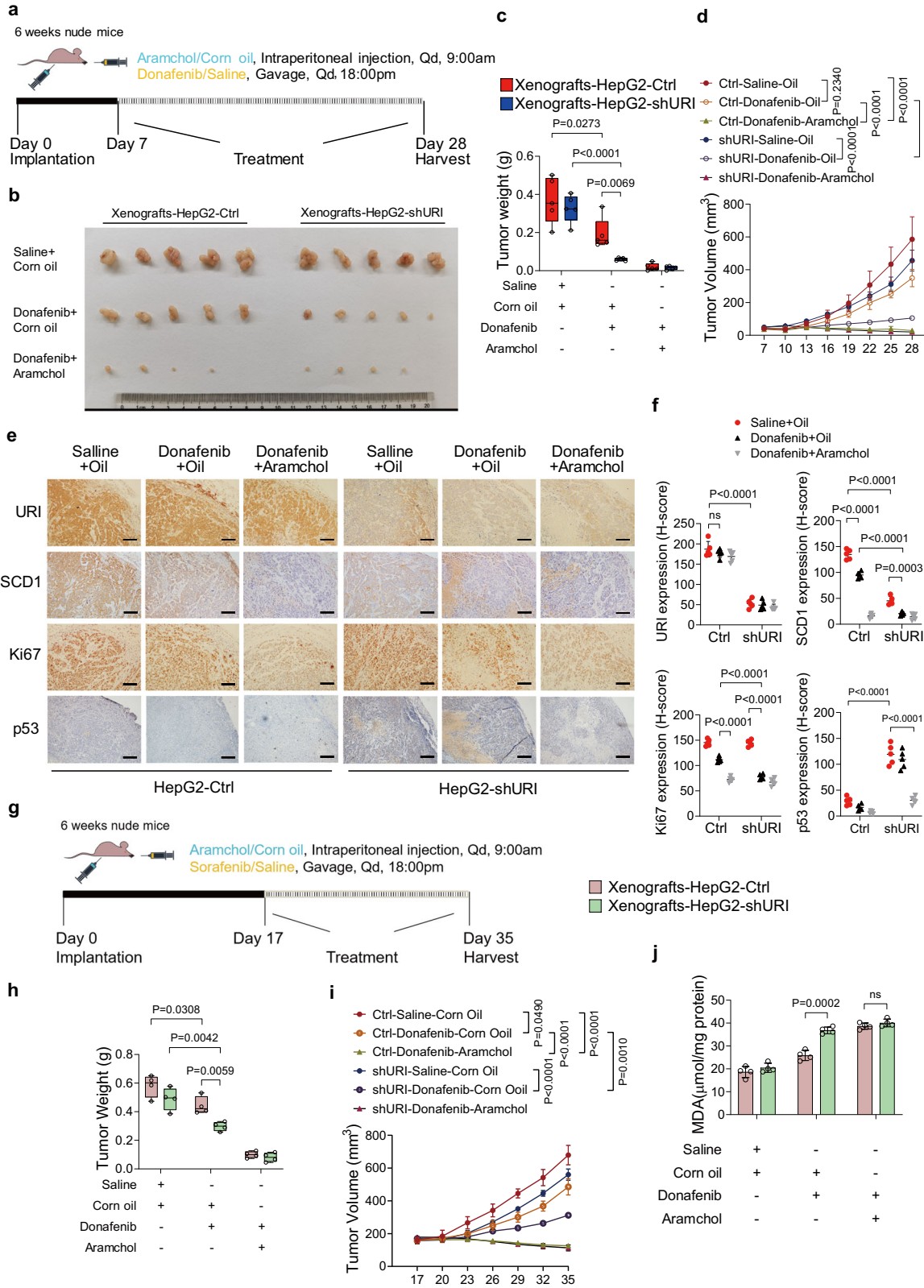

URI-p53-SCD1 axis mediates resistance of TKIs and may explain why p53-wild type HCC still showed intrinsic resistance to TKIs. Moreover, the combination therapy identified here may represent a promising strategy for approximately 41% of patients with advanced HCC who have wild-type p53 and high levels of URI/SCD1 (Fig. 9). This study provides a theoretical basis for our subsequent clinical trials.

## Methods

### Ethics declarations

All procedures involving human specimens in this study were approved by the Committee on Ethics of Medicine, Eastern Hepatobiliary Surgery Hospital (EHBH), Shanghai, China. The procedures involving animals were performed according to the

**Fig. 7 | SCD1 inhibitor aramchol enhances the efficacy of donafenib in vivo.**
**a** Schematic representation of the therapy schedule for donafenib or combination therapy. **b** Representative tumor images of each group of HepG2-Ctrl and HepG2-shURI xenografts at the end of treatment (*n* = 5 mice per group). **c** Tumor weight of each group of HepG2-Ctrl and HepG2-shURI xenografts at the end of treatment is plotted, boxplot center line, mean; box limits, upper and lower quartile; whiskers min. to max (*n* = 5 mice per group). **d** Growth curves of each group of HepG2-Ctrl and HepG2-shURI xenografts (*n* = 5 mice per group). **e** Representative immunostaining images of URI, SCD1, Ki67, and p53 in sections of xenografted tumors (*n* = 5 mice per group). Scale bar, 100 μm. **f** Quantification of IHC staining shown in (**e**) was determined by using H-score (*n* = 5 mice per group). **g** Schematic representation of the therapy schedule of donafenib or combination therapy for advanced xenografted tumors. **h** Tumor weight of each group of HepG2-Ctrl and HepG2-shURI xenografts at the end of treatment in (**g**) is plotted, boxplot center line, mean; box limits, upper and lower quartile; whiskers min. to max (*n* = 4 mice per group). **i** Growth curves of each group of HepG2-Ctrl and HepG2-shURI xenografts in (**g**) (*n* = 4 mice per group). **j** MDA assay of HepG2-Ctrl and HepG2-shURI xenografts at the end of treatment in (**g**) (*n* = 4 mice per group). Data are means ± SEM. Two-tailed unpaired *t*-test is used for the analysis of statistical significance in (**c**), (**f**), (**h**) and (**j**). Two-way ANOVA with Tukey multiple comparisons test is used for the analysis of statistical significance in (**d**) and (**i**). Source data are provided as a Source Data file.

Naval Medical University Animal Care Facility and the National Institutes of Health guidelines. All applicable institutional and governmental regulations concerning the ethical use of animals were followed.

## Patients and specimens

The 134 patients with advanced HCC in cohort A (118 males, 16 females) are from EHBH, Shanghai, China with median age of 48 (range 28–73) and treated with hepatectomy. All the 97 patients (89 males, 8 females) in cohort B are from EHBH with median age of 46 (range 17–74), and these patients were at high risk of recurrence and received sorafenib therapy after hepatectomy. The detailed clinical characteristics of Cohort A and Cohort B are provided in our Supplementary Data 4 and Supplementary Data 5, respectively. In cohort C, as mentioned in its related published article[46], the HCC patients diagnosed for the first time at EHBH and underwent hepatectomy during 2006–2013, and 119 patients whom relapsed after the surgery and not suitable for the second operation were enrolled. The enrolled patients received mono sorafenib therapy or systemic therapy containing sorafenib according to NCCN guidelines for advanced hepatobiliary cancers. The collected specimens were analyzed by targeted exome sequencing. After the default quality control and a modest filter (coverage > 150 and broadness > 60% for the targeted regions), together with excluded samples with the abnormal high mutation rates, total 80 patients (72 males, 8 females) were left with median age of 51 (range 43–57). The detailed clinical characteristics and mutation landscapes can be viewed in the published article and the dataset CRA001003 in NGDC database. The mutation status of TP53 is provided in Supplementary Data 6. In HCC_Fudan_Cohort[45], the detailed clinical characteristics of patients can be viewed in the published article[41]. Briefly, the patients were initially enrolled for CPTAC project (CHCC-HBV patients) and underwent primary curative resection from 2010 to 2014 at Zhongshan Hospital, Shanghai. The paired tumor, adjacent non-tumor liver tissues and blood samples were collected according to the CPTAC clinical sample collection procedures and the RNA-seq and WES date were collected. The patients with both WES and RNA-seq date were finally enrolled. The median age was 54 (range 20–79) and 80.6% of patients were males and 19.4% of patients were females. Recurrence-free survival (RFS) and overall survival (OS) analysis were performed using the Kaplan–Meier method. RFS was defined as the interval between the date of surgery and recurrence. OS was defined as the interval between the date of surgery and death. If recurrence was not diagnosed, patients were censored on the date of the last follow-up.

## Cell lines and cell culture

HepG2 (p53-wild), Hep3B (p53-null), Huh7 (p53-Y220C), HCT116 (p53-wild), HT29 (p53-R273H), and PLC/PRF/5 (p53-R249S) cells were purchased from Shanghai Cell resource center of Chinese Academy of Sciences. JHH1 (p53-wild) cells were kindly provided by Haojie Jin at the Shanghai Cancer Institute. HEK293T cells were purchased from the American Type Culture Collection (ATCC). All of the cell lines expect JHH-1 are maintained by the supplier and no additional authentication was performed. Short tandem repeat (STR) profiling of JHH-1 cell was tested. All cell lines were confirmed to be mycoplasma free with a mycoplasma detection kit and treated with Mycoplasma Elimination Reagent for the prevention of mycoplasma contamination. HepG2, Hep3B, Huh7, PLC/PRF/5, JHH1, and HEK293T cells were cultured in DMEM (BasialMedia, with 4.5 g/L glucose, 4mM L-glutamine, and 0.11 g/L pyruvate). HCT116 and HT29 cells were cultured in RPMI-1640 (BasialMedia, with 2 g/L glucose and 300 mg/L L-glutamine). Culture medium was supplemented with 10% Foetal Bovine Serum (Biological Industries, 04-001-1ACS), penicillin (10000 U/mL)/streptomycin (10000ug/mL), and amphotericin B (25 μg/mL) (BasialMedia). Mycoplasma Elimination Reagent (InvivoGen, 25 mg/mL) was used for the last 3 days before freezing. Fresh cells were recovered every month to maintain cell viability and a low number of cell lines passage. Cells were maintained at 37 °C in an atmosphere of humidified air containing 5% $CO_2$. Cells were routinely passaged every 2 days and not allowed to grow to confluence.

## Culture of human HCC derived organoids

Fresh tissue samples were minced and incubated in digestion buffer (DMEM (Gibco) with 0.1 mg/mL DNase I (Sigma), 4 mg/mL collagenase D (Roche), $2 \times 10^{-6}$ M Y27632 (Sigma-Aldrich), 100 μg/mL Primocin (InvivoGen)) for 30–90 min at 37 °C. The suspension was filtered through a cell strainer (100 μm) and centrifuged at $100 \times g$ for 5 min. The pellet was washed twice in cold Advanced DMEM/F12 (GIBCO, USA), and then mixed with Matrigel (BD Transduction Laboratories, USA). 150 μL cold Matrigel mixed with 200 μL cell suspension was cultured in 6 6-well suspension culture plate at 37 °C for 30 min. After gelation, 2 mL medium was added to each well. The organoid culture medium consisted of Advanced DMEM/F12 supplemented with 1% penicillin/streptomycin, 1% GlutaMAX-I, 100 μg/mL Primocin, $10 \times 10^{-3}$ M HEPES, 1:50 B27 supplement (without vitamin A), $1.25 \times 10^{-3}$ M N-acetyl-l-cysteine, 1 ng/mL recombinant human FGF-basic, 50 ng/mL mouse recombinant EGF, 100 ng/mL recombinant human FGF10, $10 \times 10^{-6}$ M forskolin, 25 ng/mL recombinant human HGF, $5 \times 10^{-6}$ M A8301, $10 \times 10^{-6}$ M Y27632, 10%, vol/vol Rspo-1 conditioned medium, 30%, vol/vol Wnt3a-conditioned medium, and 5%, vol/vol Noggin conditioned media.

## Real-time PCR analysis

Total RNA was isolated from cells using TRIzol Reagent (ThermoFisher, 15596-018) and RNeasy Mini Kit (Qiagen, 74104). For RT-PCR, mRNA was reverse transcribed into cDNA using the M-MLV system (Promega, M1701). Real-time PCR was performed on LightCycler 480 II system using a SYBR Green PCR Master Mix (Roche), with 18S as a reference control. Primers are listed in Supplementary Data 8.

## Lipid peroxidation assay

C-11 BODIPY assay. Cells were resuspended in 500μl PBS containing 20 mM C11-BODIPY 581/591 (ABclonal, RM02821) and incubated for 1 h at 37 °C in a cell culture incubator. The signals from both non-oxidized C11 (wave length > 580 nm) and oxidized C11 (wave length 505–550 nm) were monitored. The data were normalized to control samples as shown by relative lipid peroxidation.

Liperfluo assay. The Liperfluo (Dojindo, L248) was used to evaluate cell lipid peroxidation according to the manufacturer's

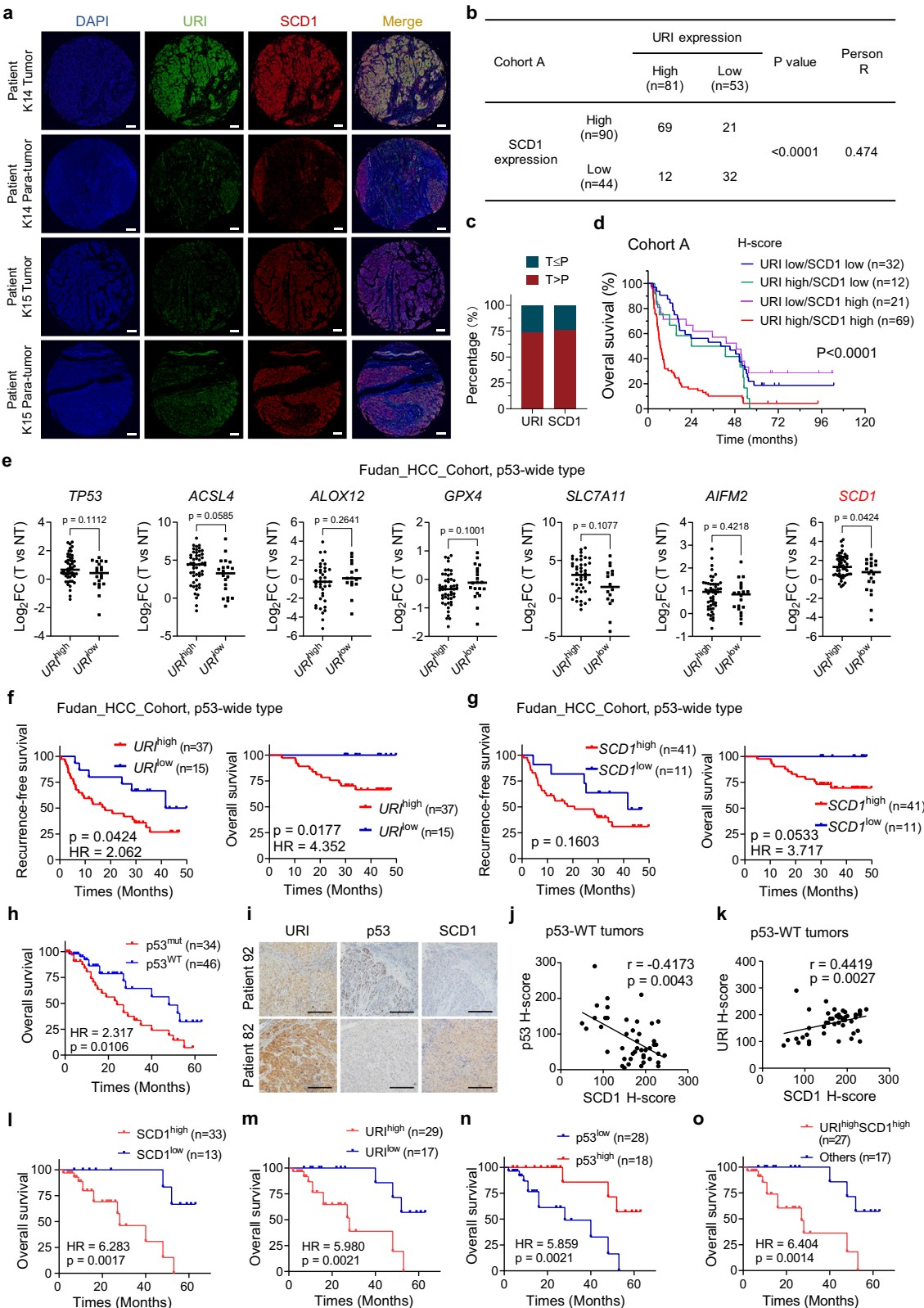

instructions. In brief, cells were treated with drugs for 48 h and incubated with 5 μM Liperfluo for 30 min at 37 °C and then detected by flow cytometry at 488 nm excitation and 550 nm emission.

MDA assay. For MDA assay, cells were plated in 6-well plates at a density of $6 \times 10^5$ cells/well and then treated with drugs for 48 h. The intracellular levels of MDA were detected by MDA Assay Kit (Dojindo, M496).

**Cell proliferation, colony formation, and viability assay**

Cell proliferation was measured using a CCK-8 Cell Counting Kit (Vazyme, A311) according to the manufacturer's instructions. For colony formation assay, cells were plated in 6-well plates at a density of 3000–5000 cells/well and treated with drugs 24 h later. Colonies were stained with crystal violet after 12–14 days and then counted by using Image J for quantification. For cell viability assessment, cells were

**Fig. 8 | URI combination with SCD1 is associated with poor survival and sorafenib-resistance in advanced HCC patients. a** Representative images of mIHC/IF on samples from cohort A with BCLC B/C stage HCC (*n* = 134 patients). URI, green; SCD1, red. Nuclei, DAPI (blue). Para-tumor, the adjacent normal-like tissue. Scale bar, 200 µm. **b** Pearson's correlation between URI and SCD1 expression in HCC tissues in cohort A by two-tailed test (*n* = 134 patients). **c** Percentage of samples with high expression of URI and SCD1 in cohort A. T, tumor; P, para-tumor (*n* = 134 patients). **d** Survival analysis of four subgroups (URI low/SCD1 low, URI high/SCD1 low, URI low/SCD1 high and URI high/SCD1 high) from cohort A (*n* = 134 patients). **e** The transcriptional status of certain genes between *URI*high versus *URI*low patients with wild-type p53 from Fudan_HCC_Cohort[45], the transcriptional levels were showed as log2 values of gene FPKM ratios between tumor (T) and paired non-tumor (NT), and the *URI*high patients had higher *URI* levels in tumors than their paired non-tumor tissues, while the *URI*low patients had lower tumoral *URI* levels than non-tumor tissues. **f** Recurrence free survival analysis and overall survival analysis of *URI*high group and *URI*low group in Fudan_HCC_Cohort with wild-type p53 (*n* = 52 patients). **g** Recurrence free survival analysis and overall survival analysis of

*SCD1*high and *SCD1*low group in Fudan_HCC_Cohort with wild-type p53 (*n* = 52 patients), the *SCD1*high patients had higher *SCD1* levels in tumors than their paired non-tumor tissues, while the *SCD1*low patients had lower tumoral *SCD1* levels than non-tumor tissues. **h** Survival analysis of p53mut and p53wt group in sorafenib-treated cohort C[46] (*n* = 80 patients). **i** Representative IHC staining of URI, p53 and SCD1 in patients with p53WT status from cohort C. Scale bar, 200 µm. **j** Spearman's correlation between SCD1 H-score and p53 H-score in p53-wild type (p53-WT) tumors from cohort C (determined by two-tailed test) (*n* = 46 patients). **k** Spearman's correlation between SCD1 H-score and URI H-score in p53-WT tumors from cohort C (determined by two-tailed test) (*n* = 46 patients). **l** Survival analysis of SCD1high and SCD1low patients in p53-WT tumors from cohort C (*n* = 46 patients). **m**, Survival analysis of URIhigh and URIlow patients in p53-WT tumors from cohort C (*n* = 46 patients). **n** Survival analysis of p53high and p53low patients in p53-WT tumors from cohort C (*n* = 46 patients). **o** Survival analysis of URIhigh/SCD1high and other groups in p53-WT tumors from cohort C (*n* = 46 patients). Data are means ± SEM. HR: hazard ratio; Two-sided log-rank test (**d**, **f**–**h**, **l**–**o**), or two-tailed Mann–Whitney test (**e**). Source data are provided as a Source Data file.

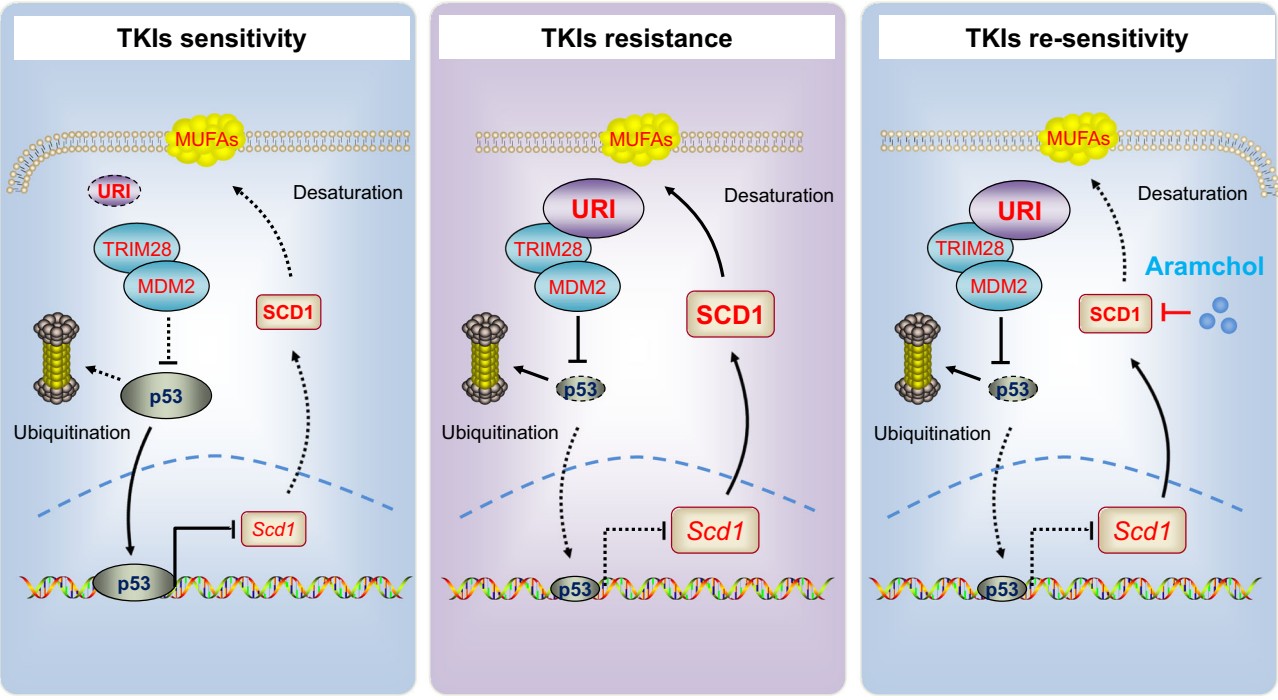

**Fig. 9 | Model for the URI-p53-SCD1 axis mediating TKIs resistance and administration of aramchol re-sensitizing liver cancer cells to TKIs.**

cultured in 96-well plates and tested using the CellTiter-Glo luminescent cell viability assay (Promega, G7571).

### Flow cytometry
Cells (30,000–50,000 cells/well) were seeded in six-well plates and treated with drugs for 48 h. For tumor cell death detection, cells were resuspended in 500 µl phosphate buffer saline (PBS) and incubated with Propidium Iodide (PI) (Sigma-Aldrich, P4170) for 30 min and then analyzed by the flow cytometry performed with LSRFortessa X-20 flow cytometer (BD Biosciences).

### Western blotting, co-immunoprecipitation, and antibodies
Cells were lysed in RIPA Lysis Buffer (Beyotime, P0013B) on ice and centrifuged at 13,500 × *g* 4 °C for 15 min. Protein concentrations were measured using a BCA protein assay kit (Thermo Scientific, 23225). Blotting was performed following the standard procedures and detected using an Odyssey fluorescence scanner (Li-Cor, Lincoln, Neb). Antibodies used are URI (1:1000, Proteintech, 11277-1-AP,), SREBP1 (1:200, Santa Cruz, sc-13551), SCD1 (1:1000, ABclonal, A16429), FASN (1:1000, ABclonal, A19050), FADS2 (1:1000, ABclonal, A10270), GPX4

(1:1000, ABclonal, A11243), GCLM (1:1000, ABclonal, A11444), GCLC (1:1000, ABclonal, A4499), SLC7A11 (1:1000, Proteintech, 26864-1-AP), ACSL4 (1:1000, Proteintech, 22401-1-AP), MDM2 (1:1000, ABclonal, A13327), Myc-Tag (1:1000, ABclonal, AE010), Flag-Tag (1:1000, ABclonal, AE005), His-Tag (1:1000, ABclonal, AE003), HA-Tag (1:1000, ABclonal, AE008), ACC (1:1000, CST, 3676), TRIM28 (1:1000, Proteintech, 15202-1-AP,), P53 (1:1000, Proteintech, 10442-1-AP), P21 (1:1000, Proteintech, 10355-1-AP), USP5 (1:1000, ABclonal, A4202), USP7 (1:1000, ABclonal, A13564), USP14 (1:1000, ABclonal, A19589), Actin (1:5000, Proteintech, 66009-1-ig), GAPDH (1:5000, ABclonal, AC001).

For co-immunoprecipitation assay, cell lysates were prepared with IP lysis buffer (Beyotime, P0013) and incubated with antibodies overnight at 4 °C, followed by the addition of Mag 25 K/Protein A/G (Enriching, LM220622) for another 4 h. Immune complexes were isolated by centrifugation and proteins were eluted into loading buffer, followed by western blotting.

### Luciferase reporter assay
To assess SCD1 gene promotor activity, cells were co-transfected with SCD1 luciferase reporter plasmid, with Renilla control reporter as an

internal control. After 48 h, cells were lysed in a Passive Lysis 5X Buffer (Promega, E1941), and the enzymatic activity of luciferase were measured using a Dual-Luciferase Assay kit (Vazyme, DL101-01), according to the manufacturer's protocol. To construct SCD1 luciferase reporter plasmid, the 1547-bp sequence above the TSS site of SCD1 was inserted in a pGL3-Basic vector using the specific enzymes at KpnI and XhoI sites.

## Chromatin immunoprecipitation (ChIP) assay

ChIP assay was performed using the ChIP Assay Kit (Beyotime, P2078). Briefly, $1 \times 10^7$ cells were cross-linked with formaldehyde/PBS for 10 min at 4 °C, and then added glycine solution, reaction at room temperature for 5 min. Subsequently, samples were sonicated in lysis buffer to obtain 200–1000 bp DNA fragments to be immunoprecipitated with 2 μg of mouse SREBP1 (Santa Cruz, sc13551), mouse P53 (Santa Cruz, sc126) or mouse IgG antibodies. The primer sequences specific to the promoter region of gene are listed in Supplementary Data 8.

## Tumor xenograft experiment

All mice were treated according to protocols approved by the Naval Medical University Animal Care Facility and the National Institutes of Health guidelines. Pathogen-free male athymic nude mice (6-week-old, D000521) were purchased from the GemPharmatech Co., Ltd (Jiangsu, China). All the animals were housed in a specific pathogen-free (SPF) environment on a 12 h light/dark cycle at temperature 20–25 °C and humidity 50–60%. For cell line derived xenografts (CDX) models, ~1 × 10^7 liver cancer cells in 0.1 mL PBS were injected subcutaneously into the right flank of 6-week-old male nude mice. Tumor volume was measured using a caliper every 3 days and calculated using the modified ellipsoidal formula: tumor volume $(mm^3) = (length \times width^2) / 2$. The maximal tumor size/burden permitted by our ethics committee review board was 2000 mm³. We confirm that none of the mice included in this study exceeded this limit. After tumor establishment, mice were randomly assigned to once per day treatment. Mice were sacrificed by cervical dislocation when the volume reached 1–1.5 cm³ and tumor tissues were harvested for further study.

## Immunohistochemistry and scoring

Ten percent formalin-fixed paraffin-embedded tissues were sectioned (4 μm) and stained with hematoxylin and eosin (H&E) for histological analysis or used for immunohistochemistry (IHC). For IHC, endogenous peroxidases were inactivated by 3% hydrogen peroxide and nonspecific signals were blocked by 1% BSA. Sections were incubated with primary antibody at 4 °C overnight, HRP-conjugated secondary antibody at 37 °C for 1 h, and subsequently stained with DAB substrate. Counterstaining was performed with hematoxylin, and mounted with a mounting medium. Antibodies are listed in Supplementary Data 7.

*H-score* assay was performed, ranging from 0 to 300. URI and SCD1 staining was scored according to four categories: 0 for 'no staining', 1+ for 'light staining', 2+ for 'intermediate staining'; and 3+ for 'dark staining'. The percentage of cells at different staining intensities was determined by visual assessment, with the score calculated using the formula 1 × (% of 1+ cells) + 2 × (% of 2+ cells) + 3 × (% of 3+ cells). The outcome-based discriminatory threshold IHC *H-score* for this analysis was set at 200 and samples were then classified as either low (*H*-score < 200) or high (*H*-score ≥ 200) for URI and SCD1 protein expression.

## Multiplex immunohistochemistry/immunofluorescence (mIHC/IF) staining

To investigate the expression of URI and SCD1 in patient tissues, mIHC/IF was conducted using a tissue microarray. Briefly, endogenous peroxidases were blocked by 3% hydrogen peroxide and nonspecific signals were blocked by 1% BSA. The slides were immersed in Tris-EDTA buffer to perform heat-induced antigen retrieval. The slides were then incubated with anti-URI (Proteintech, 11277-1-AP), anti-SCD1 (Abcam, ab236868) antibodies. Signal detection were performed using a TSA kit (AKOYA) and nuclei were counterstained with DAPI. Slides were imaged and scanned using a slice scanner (Pannoramic MIDI: 3Dhistech, Hungary) and images were analyzed via HALO 2.0 Area Quantification algorithm (Indica Labs; Corrales, NM), at Nanjing Freethinking Biotechnology Co., Ltd. (China).

## RNA sequencing

Total RNA was isolated using a RNeasy mini kit (Qiagen, Germany) from three biological replicates. TruSeq Stranded Total RNA Sample Preparation kit (Illumina, USA) was used to prepare the strand-specific libraries following the manufacturer's instructions. Briefly, using the oligo (dT) beads to enrich mRNA. Following purification, the mRNA is fragmented into small pieces using divalent cations under 86 °C for 6 min. The cleaved RNA fragments are copied into first strand cDNA using reverse transcriptase and random primers. This is followed by second strand cDNA synthesis using DNA Polymerase I and RNase H. These cDNA fragments then go through an end repair process, the addition of a single 'A' base, and then ligation of the adapters. Products are then purified and enriched with PCR to create the final cDNA library. Purified libraries were quantified by Qubit 2.0 Fluorometer (Life Technologies, USA) and validated by Agilent 2100 bioanalyzer (Agilent Technologies, USA) to confirm the insert size and calculate the mole concentration. Cluster was generated by cBot with the library diluted to 10 pM and then were sequenced on the Illumina NovaSeq 6000 (Illumina, USA). The library construction and sequencing was performed at Shanghai Biotechnology Corporation.

For each sample, 33–95 M RNA-seq clean reads were obtained that mapped to homo sapiens using HISAT2. Sequencing read counts were calculated using Stringtie (v.1.3.0). Then expression levels from different samples were normalized by the Trimmed Mean of M values (TMM) method. The normalized expression levels of different samples were converted to FPKM (Fragments Per Kilobase of transcript per Million mapped fragments). The edgeR package of R was used to analyze the difference between intergroup gene expression. The *P*-value threshold is determined by controlling the FDR (False Discovery Rate) with the Benjamini algorithm. The corrected *P*-value is called the q-value. Differentially expressed genes (DEGs) were defined as transcripts with a fold change in expression level (according to the FPKM value) >2.0 and a *q*-value < 0.05. Kyoto Encyclopedia of Genes and Genomes (KEGG) pathway and GO enrichment analysis was performed with the clusterProfiler package of R and the enrichment criteria including a *q*-value < 0.05. Heatmaps of specific genes were generated using the pheatmap package of R. Gene set enrichment analysis (GSEA) was performed using GSEA software.

## CUT&Tag library construction

HepG2 cells were treated with 5 μM Nutlin-3 for 4 h. Dead cells were identified by trypan blue staining. Ensure that the cell viability is greater than 95% before detection. Hyperactive™ In-Situ ChIP Library Prep Kit (TD901-TD902, Vazyme Biotech) for Illumina was used to perform CUT&Tag assay[53]. Briefly, concanavalin A-coated magnetic beads (ConA beads) were added to resuspended cells and incubated at room temperature to bound cells. Non-ionic detergent Digitonin was used to permeate cell membrane. Then, mouse p53 antibody (sc126, Santa), secondary antibody and the Hyperactive pA-Tn5 Transposase were incubated with the cells that were bounded by ConA beads in order. Therefore, the Hyperactive pA-Tn5 Transposase can exactly cut off the DNA fragments that were bound with target protein. In addition, the cut DNA fragments can be ligated with P5 and P7 adaptors by Tn5 transposase and the libraries were amplified by PCR with the P5

and P7 primers. The purified PCR products were evaluated using the Agilent 2100 Bioanalyzer (Agilent Technologies, Santa Clara, CA, USA). Finally, these libraries were sequenced on the Illumina NovaSeq6000 platform and 150 bp paired-end reads were generated for the following analysis by Shanghai OEbiotech company.

## Lipidomics by liquid chromatography coupled to mass spectrometry (LC-MS)

Lipid extraction and mass spectrometry-based lipid detection were performed by BioNovaGene (Suzhou, China). Briefly, take an appropriate amount of sample in a 2 mL EP tube, add 750 μL of Chloroform methanol mixed solution (2:1), vortex for 30 s. Add 2 steel beads, put them into the tissue grinder, and grind at 50 Hz for 60 s, repeated 2 times. Put on the ice for 40 min, add 190 μL $H_2O$, vortex for 30 s, put on the ice for 10 min. Centrifuged at $13,500 \times g$ for 5 min at room temperature and transfer 300 μL lower layer fluid into a new centrifuge tube. Then add 500 μL of Chloroform methanol mixed solution (2:1), vortex for 30 s. Centrifuged at $13,500 \times g$ for 5 min at room temperature and transfer 400 μL lower layer fluid into the same centrifuge tube above. Samples were concentrated to dry in a vacuum. Dissolve samples with 200 μL isopropannol, and the supernatant was filtered through 0.22 μm membrane to obtain the prepared samples for LC-MS.

Chromatographic separation was accomplished with an ACQUITY UPLC® BEH C18 (100 × 2.1 mm, 1.7 μm, Waters) column maintained at 50 °C. The temperature of the autosampler was 8 °C. Gradient elution of analytes was carried out with acetonitrile:water = 60:40(0.1%formic acid +10 mM ammonium formate)(C) and isopropanol: acetonitrile = 90:10(0.1%formic acid +10 mM ammonium formate)(D) at a flow rate of 0.25 mL/min. Injection of 2 μL of each sample was done after equilibration. An increasing linear gradient of solvent C (v/v) was used as follows: 0–5 min, 70%–57% C; 5–5.1 min, 57%–50% C; 5.1–14 min, 50%–30% C; 14–14.1 min, 30% C; 14.1–21 min, 30%–1% C; 21–24 min, 1% C; 24–24.1 min, 1%–70% C; 24.1–28 min, 70% C. The electrospray tandom mass spectrometry (ESI-MSn) experiments were used with the spray voltage of 3.5 and −2.5 kV in positive and negative modes, respectively. Sheath gas and auxiliary gas were set at 30 and 10 arbitrary units, respectively. The capillary temperature was 325 °C. The Orbitrap analyzer scanned over a mass range of $m/z$ 150–2000 for full scan at a mass resolution of 35,000. Data dependent acquisition (DDA) MS/MS experiments were performed with HCD scan. The normalized collision energy was 30 eV. Dynamic exclusion was implemented to remove some unnecessary information in MS/MS spectra.

## Targeted medium- and long-chain fatty acids quantitation

LC-MS analysis of medium- and long-chain fatty acids in HepG2-Ctrl and HepG-shURI cells was performed by Shanghai Sensichip Infotech Co. (Shanghai, China). Briefly, all the samples were mixed with 600 μL 50% acetonitrile, vortex for 1 min, and centrifuged at $13,500 \times g$, 4 °C for 15 min. Then, 200 μl of supernatant was added with 100 μL 200 mM 3-NPH and 100 μL 120 mM EDC (6% pyridine, 400 ng/mL acetic acid-D3) and was incubated at 40 °C for 1 min. Samples were centrifuged at $13,500 \times g$, 4 °C for 15 min and their supernatants were transferred to tubes for LC-MS analysis. LC-MS data was acquired on AB SCIEX 5500 QQQ (Applied Biosystems, Foster City, CA, USA) mass spectrometer coupled with high-performance liquid chromatography (HPLC) system ACQUITY UPLC. (Waters, Milford, MA, USA). The column for chromatographic separation was a ACQUITY UPLC BEH Amide Column (2.1 × 100 mm, 1.7 μm). For determination of relative metabolite abundances, the total abundances were normalized to an internal standard (acetic acid-D3) and the weight for cell extracts.

## Proteins LC-MS/MS analysis

For in-gel tryptic digestion, gel pieces were destained in 50 mM $NH_4HCO_3$ in 50% acetonitrile. Gel pieces were dehydrated with 100 μL of 100% acetonitrile for 5 min, the liquid removed, and the gel pieces rehydrated in 10 mM dithiothreitol and incubated at 37 °C for 60 min. Gel pieces were again dehydrated in 100% acetonitrile, liquid was removed and gel pieces were rehydrated with 55 mM iodoacetamide. Samples were incubated at room temperature, in the dark for 45 min. Gel pieces were washed with 50 mM $NH_4HCO_3$ and dehydrated with 100% acetonitrile. Gel pieces were rehydrated with 10 ng/μL trypsin resuspended in 50 mM $NH_4HCO_3$ on ice for 1 h. Excess liquid was removed and gel pieces were digested with trypsin at 37 °C overnight. Peptides were extracted with 50% acetonitrile/5% formic acid, followed by 100% acetonitrile. Peptides were dried to completion and resuspended in 2% acetonitrile/0.1% formic acid.

**LC-MS/MS analysis.** The tryptic peptides were dissolved in 0.1% formic acid (solvent A), directly loaded onto a home-made reversed-phase analytical column (15-cm length, 75 μm i.d.). The gradient was comprised of an increase from 6 to 23% solvent B (0.1% formic acid in 98% acetonitrile) over 16 min, 23 to 35% in 8 min and climbing to 80% in 3 min then holding at 80% for the last 3 min, all at a constant flow rate of 400 nl/min on an EASY-nLC 1000 UPLC system.

The peptides were subjected to NSI source followed by tandem mass spectrometry (MS/MS) in Q ExactiveTM Plus (Thermo) coupled online to the UPLC. The electrospray voltage applied was 2.0 kV. The m/z scan range was 350–1800 for full scan, and intact peptides were detected in the Orbitrap at a resolution of 70,000. Peptides were then selected for MS/MS using NCE setting as 28 and the fragments were detected in the Orbitrap at a resolution of 17,500. A data-dependent procedure that alternated between one MS scan followed by 20 MS/MS scans with 15.0 s dynamic exclusion. Automatic gain control (AGC) was set at 5E4.

The resulting MS/MS data were processed using Proteome Discoverer 1.3. Trypsin/P (or other enzymes if any) was specified as cleavage enzyme allowing up to 2 missing cleavages. Mass error was set to 10 ppm for precursor ions and 0.02 Da for fragment ions. Carbamidomethyl on Cys were specified as fixed modification and oxidation on Met was specified as variable modification. Peptide confidence was set at high, and peptide ion score was set > 20.

## DNA electrophoresis

10 × TAE buffer were diluted into 1 × TAE for working concentration. 2% agarose gel was made using 1 × TAE buffer with additional 10μl SolarRed (10,000×, Solarbio, G5560). The separation of DNA fragments were achieved in 2% agarose gels in 1 × TAE buffer using a DNA marker (Vazyme, DL5000) at 120 V for 25 min.

## Recombinant Myc-URI and His-TRIM28 production

The recombinant Myc-URI and His-TRIM28 were obtained from Ata-Genix Laboratories Co. (Wuhan, China). Briefly, the *URI* and *TRIM28* cDNA was obtained from the National Center of Biotechnology Information. The cDNA was cloned in a plasmidic expression vector (pET28b) harboring a histidine tag. After DNA sequence verification, the plasmid was transferred into Escherichia coli. The bacteria starter was obtained by incubation at 37 °C for 4 h. Recombinant protein was obtained using a Ni-NTA Superflow column (Qiagen, Hilden, Germany).

## Statistical analyses

All the statistical analyses were performed using GraphPad Prism8 software (GraphPad Software, Inc.). Two-tailed $t$-test and Mann–Whitney test were used to compare shURI vs. Ctrl groups. ANOVA models were used to compare continuous outcomes across multiple experimental groups. Kaplan- Meier analysis with log-rank tests was used to determine disease-free survival (DFS) and overall survival (OS). The data represent mean values of at least three independent experiments. Error bars represent mean ± SEM or median ± SEM, as appropriate. Statistical significance was set at $p < 0.05$.

**Reporting summary**

Further information on research design is available in the Nature Portfolio Reporting Summary linked to this article.

## Data availability

The data of CUT&Tag and RNA-seq of HCC cells generated in this study have been deposited in the National Genomics Data Center under accession code HRA003798 for CUT&Tag and HRA003799 for RNA-seq. The mass spectrometry proteomics data have been deposited to the ProteomeXchange Consortium via the PRIDE[54] partner repository with the dataset identifier PXD045407. The publicly available data of WES and transcriptome sequencing of Fudan_HCC_Cohort[45] used in this study can be viewed in NODE by accession OEP000321. The publicly available sequence data of Sorafenib-HCC cohort[46] (Cohort C) used in this study can be viewed in NGDC with the accession number CRA001003. The lipidomic data have been deposited in figshare (https://doi.org/10.6084/m9.figshare.23690010). The public datasets of GSE96793[27], GSE96794[27] and GSE121153[28] were employed. All remaining data are available in the Article, Supplementary and Source Data files. Source data are provided with this paper.

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

## Acknowledgements

We thank Dan Cao for preparing of paraffin encrusting; Yu-yao Zhu for sectioning of paraffin encrusting; Hao-jie Jin for providing JHH1 cell lines; Lei Chen and Yan-jing Zhu for their kind guidance on PDOs experiments. This work was supported from the National Natural Science Foundation of China (No. 81802311; 82172895; 81874182; 82073228), the Youth Program from Shanghai Municipal Health Commission (No. 20184Y0125), the Clinical Research Plan of Shanghai Hospital Development Center (No. SHDC2020CR2011A, SHDC2020CR3031B, SHDC2020CR5007), Natural Science Foundation of Shanghai (20ZR1470000), the National Key R&D Program of China (2022YFC2503704) and Open Project of State Key Laboratory of Systems Medicine for Cancer (KF2125-93).

## Author contributions

H.W., L.D., and L.W. conceived and designed the study; Z.D., Y.P., T.S., T.J., Y.L., C.Y., S.P., X.C., Y.W., X.f.F., M.X., M.P., Y.C., and X.L. performed the experiments; Z.D., Y.P., and T.S. analyzed the data. Z.D., Y.P., T.S., and L.D. prepared the manuscript; All authors read and approved the final manuscript.

## Competing interests

The authors declared no competing interests.
