## [Peer Review File · Nature Communications]

Reviewers' Comments:

Reviewer #1:

Remarks to the Author:

By using RNA-seq and lipidomic analysis in different cell lines, human samples and xenograft models, the manuscript by Ding et al aims to explore mechanisms behind resistance to tyrosine kinase inhibitors (TKIs) for treatment of advanced hepatocellular carcinoma (HCC). Authors identify SCD1 a key enzyme involved in lipid metabolism and which inhibition could be combined with TKIs to efficiently treat HCC patients with wild-type p53. To reach this conclusion, authors demonstrated that an increase of the prefoldin URI is linked to TKIs-induced ferroptosis. Authors demonstrate that URI interacts with TRIM28, leading to proteasome-mediated p53 degradation. The lack of p53 prevents the transcriptional repression of Scd1. Accordingly, SCD1 decreases TKIs-induced ferroptosis and enhances resistance of cancer cell to TKIs by increasing the ratio of monounsaturated fatty acids (MUFAs). Inhibition of URI-p53-SCD1 axis will enhance ferroptosis in patients provoking higher efficiency of TKIs in patients with HCC.

This is a very impressive and well conducted study. The work is truly extensive and solid with clear data and strong evidences. The paper is linear in the experimental procedure. Yet, there are still some points that authors have to consider prior publication in Nature communications.

Major points

- Since the mutation of p53 can be recurrent in some HCC patients, authors should correlate the incidence of those patients in order to show resistance of TKIs. Indeed, according to their results, authors should show data with less resistance to TKIs. Furthermore, how is the p53 mutation status affecting the survival plot in URI or SCD1 patients? Can authors provide this plot?
- The use of liver organoids from HCC patients could be a key experiment for reinforcing the results of the authors. Authors should collect HCC samples from patients, generate liver organoids and treat them with or without inhibitors of SCD1, shURI and sorafenib to confirm their results.
- Identification of p53 target signature by RNA-seq data analysis from previous published work in URI overexpressing mice (Accession number GSE48654) could strengthen the results. Additionally other previous models for HCC could be checked in this regard.

Minor points

- In general, font size should be increased in all figures
- There are some typo mistakes along the manuscript that should be corrected as in line 78 "metabolism"
- Authors could check the expression of SCD1 in figure 5a
- Authors should include the control of flag-SCD1 alone in figure 3h
- Authors could strengthen their claims by checking the altered pathways in publically available data from URI overexpressing mice
- p values are missing in GO table in Extended Data Figure 3g. Authors added them.
- Number of patients with URI low SCD1 low is missing in Extended Data figure 10b, which in fact is 25.
- It is not clear if the complete list of URI binding candidate proteins identified by LC-MS is provided by authors
- Authors should include n points in Figure 1f
- It is unclear the ubiquitinated SCD1 state in Extended Data Figure 5b. Could authors have swapped the ubiquitinated SCD1?
- Legend of Figure 1a should be modified, since it says "the triangle size indicates" but there are no triangles in the figure.
- Specific information should be included in Figure 7 h and g such as the time scale and months

Reviewer #2:

Remarks to the Author:

Tyrosine kinase inhibitors (TKIs) represent a type of promising drugs in hepatocellular carcinoma (HCC) treatment, while the resistance to them is a vital bottleneck to overcome. In this paper by Ding et al, authors aimed to study the role of unconventional prefoldin RPB5 interactor (URI) in HCC. They found that URI could enhance the resistance to TKIs in HCC by reprogramming the SCD1-related lipid metabolism. This endows HCC more resistant to TKIs-induced ferroptosis. Then, authors discovered that URI-mediated SCD1 upregulation is p53 dependent. They also proved that SCD1 is a p53 repressive target gene. Next, they revealed that URI could bind TRIM28 to promote the ubiquitination and degradation of p53. Finally, they showed that combination of SCD1 inhibitor with TKI has synergic effect in HCC treatment. Although this study provides some interesting findings, several critical issues need to be addressed.

Major ones:

(1) SCD1 as a p53 target has been reported before by several other papers. TRIM28-MDM2-p53 axis is not new, too. In addition, combination therapy by using SCD1 inhibitor and TKI in cancer is also not novel. These facts may weaken the novelty of this research.

(2) About the "URI reprograms SCD1-associated lipid metabolism" section, I'm curious why URI knockdown only increase the level of saturated fatty acids but not PUFA. The level of PL-PUFA in Fig 2a and b seem to be downregulated. How about the level of peroxidized PL-PUFA? This is the direct evidence to demonstrate the effect of URI is through ferroptosis. If PUFA level can't be changed by URI, how do the author explain the enhanced lipid peroxidation when inhibiting URI or SCD1?

(3) In fig 3c-e, authors need more evidence to prove that the effects of URI and inhibitors are through affecting ferroptosis. Ferroptosis inhibitors should be used to reverse these effects. In addition, oxidized PL-PUFA level should be determined.

(4) In fig 7 and related Extended figures, all the data didn't consider the p53 status (null or mutation) in the patient samples. This may undermine the conclusion of this paper.

Minor ones:

(1) Why did the authors choose URI to investigate? The rationale should be provided. Can sorafenib treatment induce URI expression? It has been reported that sorafenib could upregulate p53 level, which is opposite to the effect of URI. Therefore, how "high" should the level of URI be that could reverse the effect of sorafenib on p53 activation? What's the percentage of HCC patients bearing WT p53 and high URI?

(2) The domains in URI and TRIM28 responsible for the binding between these two proteins need to be determined.

(3) In fig 5e, there are several p53 binding peaks in the gene body region of SCD1. Their intensities are comparable to the peak located at the promoter region. Are these gene body sites responsible for p53-mediated SCD1 repression?

(4) About the xenograft model in fig 6, authors can test the combination treatment in p53 null or mutated cells. The lipid peroxidation level in the isolated xenograft tumor need to be tested.

(5) p53-mediated ferroptosis is different from what GPX4-mediated (PMID: 35087226 and 30962574). To confirm the effect of URI/SCD1 on ferroptosis is related to p53, authors can use tert-Butyl hydroperoxide (TBH) to trigger ferroptosis in ACSL4-KO cell to test their major conclusion.

(6) In Extended Data Fig. 1b, why is SLC7A11 upregulated when knocking down URI? SLC7A11 is a suppressive target of p53.

(7) In fig 4l, I suggest the authors to use dox rather than nutlin-3a to repeat this experiment. Because the authors didn't mention or use nutlin-3a in previous figures.

(8) In fig 5b, "p53" but not "P53".

(9) In this description "URI depletion significantly decreased ubiquitination of wild-type p53 in HepG2 cells (Fig. 5d and Extended Data Fig. 6b)", Extended Data Fig. 6b should be "Extended Data Fig. 6c". In addition, IgG and p53 antibody should be noted in Extended Data Fig. 6c.

(10) In fig 6i, why did aramchol and donafenib reduce p53 level?

(11) In fig 7a, e and Extended Data Fig. 10a, the intensities of the fluorescence signals of certain panels should be enhanced. It is hard to recognize the signals.

(12) Delete the full stop "." in the end of the title and several captions in the results part.

(13) In this sentence "URI depletion significantly increased the sensitivity of JHH1 and HepG2 cells to sorafenib (Extended Data Fig. 2c), with a decreased in IC50", "decreased" should be "decrease". Or you can delete the word "in".

(14) In this sentence "Lipid metabolic reprogram is involved in cancer drug resistance", "reprogram" should be "reprogramming".

(15) In this sentence "we found that URI high-expression in HCC is associated with cancer malignant and poor survival of patients", "malignant" should be "malignancy".

(16) In this sentence "This may helpful to keep p53 levels low as has been detected in cancer cells", "may" should be "may be".

(17) In this sentence "URI-p53-SCD1 axis mediates resistance of TKIs and may explain why p53-wild type HCC still showed intrinsic resistant to TKIs", "resistant" should be "resistance".

Reviewer #3:

Remarks to the Author:

This is an interesting manuscript suggesting that sorafenib and other TKIs may be more effective in HCC when cells are sensitized by SCD1 inhibitors, the combination of which causes ferroptosis. One strength of the manuscript is that it spans the gamut of experiments done in cell culture, xenografts and human patient samples. The link between levels of URI and SCD1 are convincing.

Specific comments:

There are typos throughout the manuscript. Please fix all of these.

Results section 1

The full list of genes altered in control versus shURI HepG2 cells should be provided in a table.

Please state how many genes are related to ferroptosis and what percentage of these are affected by URI KD.

Please reference and describe the SCD1 promoter used in the luciferase construct.

Extended data 5, there are two panels labeled "b", but no panel "c".

Figure 5. There is no panel labeled "n".

Figure 6h. The p53 IHC is not possible to interpret and should be improved.

Figure 6e. There is a large effect of Donafenib alone on these tumors. In light of this fact, please explain/justify how this could be a good model to study synergy between Donafenib and Aramchol.

Figure 7a It is very difficult to see the IHC. Please make brighter.

Figure 7 d,h,f and g. The color scheme shown for the interpretation of the graphs are different than the colors on the graphs and it is difficult to interpret the data because of this.

Text- Please define para-tumor.

Given the hypothesis that URI interacts with TRIM28 to promote p53 ubiquitination and degradation involving MDM2, it would be important to determine how the levels of URI and SCD1 correlate with levels of p53 in patient samples shown in Figure 7.

Response to reviewers' comments

Reviewer #1 - HCC metabolism - (Remarks to the Author):

By using RNA-seq and lipidomic analysis in different cell lines, human samples and xenograft models, the manuscript by Ding et al aims to explore mechanisms behind resistance to tyrosine kinase inhibitors (TKIs) for treatment of advanced hepatocellular carcinoma (HCC). Authors identify SCD1 a key enzyme involved in lipid metabolism and which inhibition could be combined with TKIs to efficiently treat HCC patients with wild-type p53. To reach this conclusion, authors demonstrated that an increase of the prefoldin URI is linked to TKIs-induced ferroptosis. Authors demonstrate that URI interacts with TRIM28, leading to proteasome-mediated p53 degradation. The lack of p53 prevents the transcriptional repression of Scd1. Accordingly, SCD1 decreases TKIs-induced ferroptosis and enhances resistance of cancer cell to TKIs by increasing the ratio of monounsaturated fatty acids (MUFAs). Inhibition of URI-p53-SCD1 axis will enhance ferroptosis in patients provoking higher efficiency of TKIs in patients with HCC.

This is a very impressive and well conducted study. The work is truly extensive and solid with clear data and strong evidences. The paper is linear in the experimental procedure. Yet, there are still some points that authors have to consider prior publication in Nature communications.

Major points

(1) Since the mutation of p53 can be recurrent in some HCC patients, authors should correlate the incidence of those patients in order to show resistance of TKIs. Indeed, according to their results, authors should show data with less resistance to TKIs. Furthermore, how is the p53 mutation status affecting the survival plot in URI or SCD1 patients? Can authors provide this plot?

Response: We thank for the reviewer's vital advice. Similar comment was also given by reviewer#2 and the results were very important to strength our conclusion. Since p53 mutation is common in

tumors including HCC and the kinds of p53 mutation are complicated, we thought that only the DNA sequencing data could demonstrate the p53 status in tumors. The cohorts we enrolled in the former manuscripts were lack of the genomic sequencing data, thus the p53 status could not be determined. To address this question, we first employed a new HCC cohort enrolled by Gao et.al (Cell, PMID: 31585088), which we named “Fudan_HCC_cohort”. The results were shown in Fig. 8 and Extended Data Fig. 10. By analysis the WES and transcriptome data of this cohort, we found that in HCC patients with p53-WT status, the *SCD1* expression was lower in *URI*^{low} tumors than in *URI*^{high} tumors, while other ferroptosis-associated molecules, such as *ACSL4*, *ALOX12*, *GPX4*, *SLC7A11* and *AIFM2*, were not significantly altered. Interestingly, the *SCD1* level in p53-mutation HCC patients were comparable between *URI*^{low} and *URI*^{high} tumors. Moreover, higher *URI* or *SCD1* expression in p53-WT HCC patients were correlated with poorer clinical outcome. We did not observe this correlation in p53-mutation HCC patients. Taken together, these results demonstrated that the potential correlation between URI-SCD1 and the clinical outcome of HCC patients was existed in patients with wild type p53, but not in p53-mutation ones.

To explore the URI-SCD1 axis in sorafenib resistance, we employed our previous cohort which enrolled HCC patients with recurrent HCC, the patients were then received systemic therapy containing sorafenib (named Cohort C in the revised manuscripts, PMID: 32373219). The mutation landscape of this cohort was performed. Forty-five patients were p53-WT and one patient harbored p53 synonymous mutation, whom we also grouped into p53-WT. The protein levels of SCD1, URI and p53 were measured by immunohistochemistry. We found significant correlation between SCD1 and URI in p53-WT group, but not in p53-mutation group (Fig. 8 and Extended Data Fig. 11). Meanwhile, higher levels of SCD1 or URI were associated with worsen prognosis in p53-WT HCC patients receiving sorafenib treatment, while no significant correlation were found in p53-mutation patients. Thus, our results demonstrated the important role of URI-SCD1 axis in sorafenib resistance in p53-WT HCC patients.

(2) The use of liver organoids from HCC patients could be a key experiment for reinforcing the results of the authors. Authors should collect HCC samples from patients, generate liver organoids and treat them with or without inhibitors of SCD1, shURI and sorafenib to confirm their results.

Response: Thanks for the constructive suggestion. We had obtained four HCC patient derived organoids (HCC-PDOs), two with wild-type p53 and two with mutant p53 (Fig. 6d). Consistent with our results in cell lines and xenograft tumor models, interfering URI promoted the sensitivity of TKI drugs and elevated lipid peroxidation levels of HCC-PDOs with wild-type p53 (Fig. 6e-h). The combination of SCD1 inhibitor aramchol and deuterated sorafenib derivative donafenib also displayed promising anti-tumor effects in HCC-PDOs with wild-type p53 (Fig. 6e-g). In HCC-PDOs with mutant p53, donafenib was less effective than in HCC-PDOs with wild-type p53. The combination of donafenib and aramchol synergistically upregulated lipid peroxidation levels to a certain extent (Extended Data Fig. 8e, f). However, this combination treatment showed less effect in HCC-PDO with mutant p53 than in HCC-PDO with wild-type p53 (Extended Data Fig. 8b, c). These results imply that URI could regulate the TKI sensitivity and the combination therapy identified here may be effective in p53 wild-type HCC.

(3) Identification of p53 target signature by RNA-seq data analysis from previous published work in URI overexpressing mice (Accession number GSE48654) could strengthen the results. Additionally other previous models for HCC could be checked in this regard.

Response: Thank you very much for your kind advice. We have read this article (PMID: 25453901) and analyzed the data set (GSE48654). In their mouse model, overexpression of human URI (hURI) in mice could lead to spontaneous liver cancer by inducing DNA-damage and subsequent p53-mediated apoptosis and liver injury. This is a tumorigenesis model and requires multistep process including acquiring mutations of the driver/suppressor genes in normal or premalignant liver cells. Indeed, the authors also mentioned that all tumors displayed dramatic increases in p53 abundance and phosphorylation, and the authors had pointed that the p53 was inactivated by mutation or inappropriate folded. Thus, their results showed that hURI overexpression in normal liver cells could lead to stress-induced p53 activation and finally p53 inactivation (by mutation or other mechanisms). Following the authors' opinion, these mouse tumors might be similar with the URI^{high}p53^{mut} HCC in humans.

As shown in Additional Figure 1, we then analyzed the transcriptome data between hURI-mice and control mice for 1 week, the nonpathological timepoint that no major mutation was accumulated

and the liver cells were not transformed. We found that p53-target genes were comparable or slightly changed than control mice. However, the lipid metabolism pathways, including linoleic acid metabolism and arachidonic acid metabolism, were affected by hURI overexpression at early stage. These results suggested that hURI has a critical role in lipid metabolism reprogramming. Interestingly, we found that *Scd1* was also elevated in 1-week hURI mice than control mice, supporting the promoting role of *Scd1* by URI in liver. Then we compared the transcriptome data hURI-mice and control mice at 8 weeks, the early premalignant state of HCC. Remarkably, p53-associated pathway was significantly changed in mouse with hURI at this timepoint. Collectively, these results strongly suggest that wild type p53 was involved in hURI-mediated liver damage, then p53 was inactivated for further tumor progression.

In our study, we focused on the role of URI in inhibiting TKIs in HCC with wild-type p53. We found that URI could regulate SCD1-mediated lipid metabolism in wild type-p53 dependent manner in tumor cells, this effect was contributed to URI-mediated TKI resistance in p53-WT HCC.

Additional Figure 1. The transcriptome analysis of hURI mice. (a) The heatmap of indicated genes in one-week hURI-mice and control mice. (b) KEGG analysis of differential expressed genes (DEGs) between one-week hURI-mice and control mice. (c) KEGG analysis of DEGs between eight-week hURI-mice and control mice.

Minor point

(1) In general, font size should be increased in all figures.

Response: Thank you for your very kind advice. We have adjusted the font size to 9 or 10 point in the figures to match the size of the layout.

(2) There are some typo mistakes along the manuscript that should be corrected as in line 78 “metabolism”.

Response: We are very sorry for these spelling problems. We have corrected them sentence by sentence and again apologize for these errors.

(3) Authors could check the expression of SCD1 in figure 5a.

Response: Thank you very much for your constructive comment. We have tested SCD1 when performing this experiment in Fig. 5a. When we preparing the former manuscript, considering that the topic of Figure 5 mainly focused on the regulation of p53 protein stability by URI, we did not put the SCD1 western blot strips. Now we have added the SCD1 WB strips according to your comment in Fig. 5a.

(4) Authors should include the control of flag-SCD1 alone in figure 3h.

Response: Thank you very much for pointing out the shortcomings of this experiment. We have reperformed the experiment and added the Flag-SCD1 group in Fig. 3f (original Fig. 3h).

(5) Authors could strengthen their claims by checking the altered pathways in publically available data from URI overexpressing mice.

Response: Thank you very much for your suggestion. We analyzed transcriptomic data from hURI overexpressing mice. The major results were described in the response to the major point 3.

(6) p values are missing in GO table in Extended Data Figure 3g. Authors added them.

Response: Thank you very much for your careful review. We have added the p-values in Extended Data Fig. 3f (original Extended Data Fig. 3g).

(7) Number of patients with URI low SCD1 low is missing in Extended Data figure 10b, which in fact is 25.

Response: Thank you again for your meticulous attention to detail. We apologize for this mistake. We have added the number in Extended Data Fig. 11b (original Extended Data Fig. 10b).

(8) It is not clear if the complete list of URI binding candidate proteins identified by LC-MS is provided by authors.

Response: Thanks for your suggestion, we have added the complete list of URI binding candidate proteins identified by LC-MS to the supplementary material (Supplementary Table 3).

(9) Authors should include n points in Figure 1f

Response: Thank you for your careful review, we have added n points in Fig. 1f.

(10) It is unclear the ubiquitinated SCD1 state in Extended Data Figure 5b. Could authors have swapped the ubiquitinated SCD1?

Response: Thank you for your suggestion. We have increased the concentration of the external transfer plasmid and the concentration of the antibody. We replaced the western blot strips with clearer ones in Extended Data Fig. 5b.

(11) Legend of Figure 1a should be modified, since it says “the triangle size indicates” but there are no triangles in the figure.

Response: Thank you for pointing out our mistake, we apologize for this. We have changed the description in Figure Legend of Fig. 1a.

(12) Specific information should be included in Figure 7 h and g such as the time scale and months

Response: We have added the information in Extended Data Fig. 11d, e (original Fig. 7h, g). We would like to thank you sincerely for your guidance on our article. We apologize again for the mistakes. We have proofread and corrected each one in the light of your review.

Reviewer #2 - P53, metabolism, ferroptosis, mass-spec, RNA-seq - (Remarks to the Author):

Tyrosine kinase inhibitors (TKIs) represent a type of promising drugs in hepatocellular carcinoma (HCC) treatment, while the resistance to them is a vital bottleneck to overcome. In this paper by Ding et al, authors aimed to study the role of unconventional prefoldin RPB5 interactor (URI) in HCC. They found that URI could enhance the resistance to TKIs in HCC by reprogramming the SCD1-related lipid metabolism. This endows HCC more resistant to TKIs-induced ferroptosis. Then, authors discovered that URI-mediated SCD1 upregulation is p53 dependent. They also proved that SCD1 is a p53 repressive target gene. Next, they revealed that URI could bind TRIM28 to promote the ubiquitination and degradation of p53. Finally, they showed that combination of SCD1 inhibitor with TKI has synergic effect in HCC treatment. Although this study provides some interesting findings, several critical issues need to be addressed.

Major ones:

(1) SCD1 as a p53 target has been reported before by several other papers. TRIM28-MDM2-p53 axis is not new, too. In addition, combination therapy by using SCD1 inhibitor and TKI in cancer is also not novel. These facts may weaken the novelty of this research.

Response: We thank for the reviewer's constructive comment. In our view, the novelty of our research was based on the following findings:

(1) Our previous study and other research works had revealed that URI could act as an oncogene and potential therapeutic target in liver cancers. However, whether URI could regulate sorafenib-induced cytotoxicity in HCC was unclear. Here, by employing various HCC cell lines and patient-derived organoids (we had employed the PDOs from HCC patients with p53-WT or -mutation, as shown in Fig. 6 and Extended Data Fig. 8), we had showed that URI could promote the resistance to TKIs in a p53-SCD1 dependent manner. Moreover, according to the results of our clinical cohorts, we had found that although HCC patients with p53-wild type had a better clinical outcome than p53-mutation group, higher levels of URI in the p53-wild type group still indicated worsen prognosis (Fig. 8). However, in HCC cells and tissues with p53-mutation, we had not found the effect of URI in sorafenib-resistance (Extended Figure 10 and 11). Thus, our results suggested that

the role of URI in sorafenib resistance was relied on the function of wild type p53.

(2) By screening the ferroptosis-associated molecules in p53-WT HCC cells, we had identified SCD1 as the target molecule regulated by URI, which revealed a former unknown correlation between oncogene URI and the lipid metabolism in HCC. This correlation was further confirmed in mice once hURI were overexpressed in liver for 1 weeks (as mentioned in the response to reviewer#1). The dependence of SCD1 in sorafenib resistance in p53-WT HCC cells made it suitable as the combination target.

(3) In this work, we had discovered the URI-p53-SCD1 axis in regulating lipid metabolism and sorafenib-resistance in HCC. As the reviewer mentioned, SCD1 has been identified as the p53-repressed gene, especially in ovarian cancer (PMID: 12789273). The TRIM28-MDM2-p53 pathway was also discovered in some tumors such as lung cancer (PMID: 27834954). Here we further confirmed that the TRIM28-MDM2-p53-SCD1 pathway was also existed in HCC with p53-WT, suggesting the conserved role of this pathway in various tumors. Moreover, we had found that URI, the potential oncogene in HCC, could also employ this pathway to inhibit p53 and its-related functions, further expanded the role of URI in HCC. Although we had not tested, considering the higher expression of URI in other tumors (PMID: 30209015, 26328264, 24625985), the URI-TRIM28-p53-SCD1 axis might be general in tumors.

(4) SCD1 has been found to regulate the population of liver T-ICs via modulation of ER stress, and its inhibitor could partly overcome sorafenib-resistant in HCC (PMID: 28647567). However, their work had not considered the p53 status and they performed their experiment mainly on Huh7 and PLC/PRF/5 cells, the two cells with p53 mutation. Consistent with their results, we also found that combination of donafenib with aramchol had slight reduced tumor growth of PLC/PRF/5 cells *in vivo* (Extended Data Fig. 9). Moreover, our experiment also showed that this combination treatment could induce lipid peroxidation in p53-mut PDOs, but had little effect on the cell cytotoxicity (Extended Data Fig. 8). Notably, we had found that the combination therapy could significantly induce cell death in HCC cells and PDOs with p53-WT than the inhibitors used alone, and URI had an important role in the TKI sensitivity in p53-WT cells. The role of URI in TKI resistance had not been observed in p53-mutation samples. These results suggested that HCC patients with p53-WT could achieve more benefit from this treatment. We also used aramchol, the

clinical phase 3/4 SCD1 inhibitor, in our animal experiments. Our results might be helpful to provide some evidence for the clinical use of this combination regimen in HCC patients, especially in patients with p53-WT status.

(2) About the “URI reprograms SCD1-associated lipid metabolism” section, I’m curious why URI knockdown only increases the level of saturated fatty acids but not PUFA. The level of PL-PUFA in Fig 2a and b seem to be downregulated. How about the level of peroxidized PL-PUFA? This is the direct evidence to demonstrate the effect of URI is through ferroptosis. If PUFA level can’t be changed by URI, how do the author explain the enhanced lipid peroxidation when inhibiting URI or SCD1?

Response: We reanalyzed our lipidomic data. The PLS-DA and OPLS-DA analysis of extracted lipid features exhibited clear separation and tight clustering among the groups (Fig. 2a). By analysis the composition of free fatty acids, we found that the monounsaturated fatty acids (MUFA) exhibited a much larger decrease than PUFA in HepG2-shURI cell than its control, while the saturated fatty acids (SFA) were slightly increased in these cells (Fig. 2b-d and Extended Data Fig. 3). These results suggested that URI could lead to enhanced conversion from SFA to MUFA. Consistent with this notion, we found that the protein level of SCD1, the enzyme that catalyzes the conversion from SFA to MUFA, was significantly inhibited by URI knockdown in p53-WT HCC cells. Interestingly, according to the GSE48654 data (as mentioned in the response to reviewer#1), we found that *Scd1* transcripts were elevated in hURI-mice than control in the early-stage, together with aberrant lipid metabolism.

PL-PUFAs are susceptible to ROS and their lipid peroxidation can fuel ferroptosis cascade. On the contrary, MUFAs could suppress this process by promoting the displacement of PUFAs from plasma membrane phospholipids (PMID: 30686757). We then analyzed the lipid species of phospholipids (such as PC, PE, PI) between HepG2-shURI and HepG2-Ctrl cells. As shown in Fig. 2, there was a decreasing tendency of MUFA in phospholipids of HepG2-shURI cells than controls under steady-state. The contents of C16:0/C20:4 PL-PUFA were increased in HepG2-shURI than control cells, while the levels of PL-MUFA C16:0/C18:1 were decreased in HepG2-shURI cells (Fig. 2f). Thus, although no significant change in PUFAs was found between HepG2-shURI and control

cells, the PL-PUFA was decreased.

We then measured the lipid peroxidation levels between HCC cells by C11-BODIPY, Liperfluo and MDA analysis. We found that comparable levels of lipid peroxidation between HCC tumors with shURI or control (Fig. 3e, Extended Data Fig. 4i). However, when these cells were treated with sorafenib, higher increased levels of lipid peroxidation were observed in HCC cells with shURI than controls, together with a significant reduction of cell viability (Fig. 3, Extended Data Fig. 4). SCD1 inhibitors alone also had little effect in lipid peroxidation (Fig. 3).

Taken together, these results suggested that URI or SCD1 alone could affect the lipid composition of HCC cells, which made them more sensitive to ferroptosis inducer, such as TKIs.

(3) In fig 3c-e, authors need more evidence to prove that the effects of URI and inhibitors are through affecting ferroptosis. Ferroptosis inhibitors should be used to reverse these effects. In addition, oxidized PL-PUFA level should be determined.

Response: We thank for the reviewer's constructive comment. We had used the ferroptosis inhibitor ferrostatin-1 (Ferr-1) to investigate whether this treatment could reverse the URI and inhibitors induced cytotoxicity. As shown in Fig. 3 and Extended Data Fig. 4, Ferr-1 treatment could inhibit sorafenib-induced cell death in HepG2 and JHH1 cells by cell viability assay and colony formation assay. Meanwhile, SCD1 inhibitors A939572 and MK8245 mediated synergistic effect in sorafenib-induced cell death could also be reversed by Ferr-1 treatment. We then measured the lipid oxidization by Liperfluo staining and MDA test. When cells were treated with sorafenib, the lipid oxidization levels in all cells tested were increased, and shURI cells showed much higher levels than their controls. Combination with SCD1 inhibitors further increased the lipid oxidation contents in cells. Notably, Ferr-1 treatment could inhibit the lipid oxidation status in cells. Taken together, our results had showed that ferroptosis is the major cell death form in sorafenib-treated HCC cells and URI-SCD1 axis could regulate sorafenib-induced ferroptosis.

(4) In fig 7 and related Extended figures, all the data didn't consider the p53 status (null or mutation) in the patient samples. This may undermine the conclusion of this paper.

Response: Thanks for the constructive suggestion. This comment and the major point one of the

Reviewer#1 are very important to strengthen the conclusion of our paper. Considering that p53 mutation is frequency in tumors and its mutation form is various between patients, we decided to obtain the p53 status by their WES or target DNA sequencing data. We enrolled a HCC cohort from Gao et.al (Cell, PMID: 31585088) and named it as “Fudan_HCC_cohort”, which had complete data of WES, transcriptome and clinical information. As shown in Fig. 8 and Extended Data Fig. 10, we found that *SCD1* expression was associated with *URI* levels in p53-WT group, but not in p53-mutation group. Higher *URI* or *SCD1* expression in p53-WT HCC patients were correlated with poorer clinical outcome. We did not observe this correlation in p53-mutation HCC patients. These results suggested an important role of *URI/SCD1* in HCC progression in patients with wild type-p53.

Then we employed our previous cohort which enrolled HCC patients with recurrent HCC, the patients were then received systemic therapy containing sorafenib (PMID: 32373219). The cohort was named as cohort C and containing 45 patients with p53-WT, 1 patient with p53- synonymous mutation and 34 patients with p53-mutation (including nonsynonymous mutation, splicing and stop-gain mutation). The levels of *SCD1*, *URI* and p53 were measured by immunohistochemistry. Significant correlation between *SCD1* and *URI* was found in p53-WT group, but not in p53-mutation group (Fig. 8, Extended Data Fig. 11). Higher levels of *SCD1* or *URI* were associated with worsen prognosis in p53-WT HCC patients receiving sorafenib treatment, while no significant correlation were found in p53-mutation patients. Thus, our results demonstrated the important role of *URI-SCD1* axis in sorafenib resistance in p53-WT HCC patients

Minor ones:

(1) Why did the authors choose *URI* to investigate? The rationale should be provided. Can sorafenib treatment induce *URI* expression? It has been reported that sorafenib could upregulate p53 level, which is opposite to the effect of *URI*. Therefore, how “high” should the level of *URI* be that could reverse the effect of sorafenib on p53 activation? What’s the percentage of HCC patients bearing WT p53 and high *URI*?

Response: We thank for the kind comment. As mentioned in the major point 1, *URI* is higher expressed in most HCC tumors than their counterpart non-tumor liver tissues, and *URI* has been

considered as an oncogene and potential therapeutic target in liver cancers, thus it is worth to explore the role of URI in HCC treatment. We did not find that sorafenib treatment could upregulate URI expression in HCC cells (Extended Data Fig. 4b). In fact, it is still unknown how URI is upregulated in HCC, although HBx and c-Myc might be involved in this process (PMID: 31739577). However, we had found that the p53 and SCD1 protein levels of HCC cells were regulated in a URI-dose dependent manner in p53-WT cells (Fig. 4p). In our view, before TKI treatment, higher levels of URI in p53-WT HCC could inhibit p53-mediated SCD1 suppression, leading to the reprogramming of SCD1-related lipid metabolism, this cellular status made tumors more resistant to TKI-induced ferroptosis. Our clinical cohorts also support this notion, since higher URI levels were associated with worsen prognosis in p53-WT HCC patients. Moreover, the percentage of HCC patients bearing WT p53 and high URI was 31.5% in the Fudan_HCC_cohort and 36.25% in Cohort C, suggesting that our result may have good clinical application scenarios, which require further investigation.

(2) The domains in URI and TRIM28 responsible for the binding between these two proteins need to be determined.

Response: We thank for the constructive comment. In another manuscript which we focused on the role of URI-TRIM28 in regulation AMPK activity, we had investigated the domains responsible for URI-TRIM28 interaction. This manuscript is still in the “under review” state. In that article, we had created Flag-tagged TRIM28 and His-tagged URI truncation mutants, separately (Additional Figure 2a and 2c). TRIM28 is a RING-type ligase and its E3 ubiquitin-protein ligase activity is intrinsic to the RING domain. Co-immunoprecipitation assays revealed that N-terminal region of TRIM28 is mainly required for its interaction with URI (Additional Figure 2b). Reciprocal experiment with URI truncation mutants and TRIM28 indicated that although URI and TRIM28 interaction likely involves multiple interaction domains, C-terminal region of URI showed a stronger ability to bind to N-terminal region of TRIM28 (Additional Figure 2d). We next obtained the protein structures of URI and TRIM28 from AlphaFold Protein Structure Database. The docked pose with the highest docking scores was used for analyzing. The interface was observed between one helix of the coiled coil of TRIM28 and a fold of acidic stretch of URI (Additional Figure 2e). Taken together, this result demonstrated that the RING-domain of TRIM28 and the C-terminal region of URI were responsible

for their binding. Since this result was a part of our another manuscript, we did not put them into this revised paper. However, we would like to perform the experiments again and added the results into this paper if the reviewer considered that it is necessary.

Additional Figure 2. The interaction domain between URI and TRIM28. (a) The diagram of the TRIM28 truncation mutants. (b) Co-immunoprecipitation assay of the interaction between Flag-TRIM28 mutants and His-URI. (c) The diagram of the URI truncation mutants. (d) Co-immunoprecipitation assay of the interaction between Flag-TRIM28 and His-URI mutants. (e) The interface between URI and TRIM28.

(3) In fig 5e, there are several p53 binding peaks in the gene body region of SCD1. Their intensities are comparable to the peak located at the promoter region. Are these gene body sites responsible for p53-mediated SCD1 repression?

Response: Thanks for the kind reminding. We had checked the CUT&Tag data and found that the

major peak site in the gene body region of SCD1 was the region of Chr10: 100347233-100364826. This sequence was synthesized and inserted into pGL3-reporter plasmid. As shown in Extended Data Fig. 5g, overexpress of p53 could not alter the luciferase activity of the reporter containing this region. These results suggested that this site was not responsible for p53-mediated SCD1 repression.

(4) About the xenograft model in fig 6, authors can test the combination treatment in p53 null or mutated cells. The lipid peroxidation level in the isolated xenograft tumor need to be tested.

Response: Since the p53-null Hep3B cells were difficult to generate the xenograft model, we had used the p53-mutation PLC/PRF/5 cells to test the therapeutic effect. We found that donafenib displayed cytotoxic effect in PLC/PRF/5-derived tumors, but combination of aramchol did not further improve its effect (Extended Data Fig. 9). The MDA levels in PLC/PRF/5 tumors were increased in donafenib-treated group than controls, and again combination of aramchol did not further increase it (Extended Data Fig. 9). In p53-WT HepG2 cells, we also tested whether the combination therapy could inhibit tumor growth at late stage. The HepG2 tumors were treated with the inhibitors alone or in combination when the tumor volumes were reached about 200 mm³. The tumor growth curve and MDA assay still showed significant improvement of donafenib combination with aramchol than them used alone (Fig. 7).

(5) p53-mediated ferroptosis is different from what GPX4-mediated (PMID: 35087226 and 30962574). To confirm the effect of URI/SCD1 on ferroptosis is related to p53, authors can use tert-Butyl hydroperoxide (TBH) to trigger ferroptosis in ACSL4-KO cell to test their major conclusion.

Response: We thank for the reviewer's constructive comment. We had knockdown ACSL4 expression by siRNA transfection in JHH1 and HepG2 cells (Additional Figure 3a). We found that TBH treatment led to cell death in JHH1 and HepG2 cells, and URI-knockdown made cells more sensitivity to TBH treatment (Additional Figure 3b and 3c). The URI-mediated effect was also existed in HCC cells with ACSL4-knockdown (Additional Figure 3b and 3c).

Additional Figure 3. URI regulates cell death upon ROS stress. (a) ACSL4 expression in HCC cells were interfered by siRNA transfection. (b and c) The shURI and Ctrl JHH1 (b) and HepG2 (c) cells were transfected with siACSL4 or scramble for 24 hours, the cells were then treated with TBH (500 μ M) for 8 hours.

(6) In Extended Data Fig. 1b, why is *SLC7A11* upregulated when knocking down URI? *SLC7A11* is a suppressive target of p53.

Response: We had found that transcripts of *SLC7A11* was upregulated when URI was knockdown in HepG2 cells according to our RNA-seq data. However, we did not find any change in the protein level of *SLC7A11* between shURI and control cells (Fig. 3a, Extended Data Fig. 4c). Although *SLC7A11* has been reported as a suppressive target of p53, many other transcription factors, such as NRF2, could also regulate *SLC7A11* expression. In our view, since URI played a critical role in HCC tumorigenesis and progression, interfering URI expression might lead to broad changes in tumors besides p53 pathway. Thus, the upregulation of *SLC7A11* in shURI cells in our result may reflect the balance between the factors which promoting or inhibiting the transcription of *SLC7A11*, but this effect might be slight and did not affect its protein level. At the same time, the classical p53-target genes, such as *MDM2* and *CDKN1A*, were elevated in shURI cells than their controls (Fig. 4g), confirming the change of p53-associated pathway induced by URI knockdown.

(7) In fig 4l, I suggest the authors to use dox rather than nutlin-3a to repeat this experiment. Because the authors didn't mention or use nutlin-3a in previous figures.

Response: We thank for the kind reminding. We had repeated our experiment using Dox instead of Nutlin3a and similar result was found. The figure was replaced as Fig. 4m.

(8) In fig 5b, “p53” but not “P53”.

Response: Thanks for the kind reminding. We are sorry for our mistake and we have corrected this error.

(9) In this description “URI depletion significantly decreased ubiquitination of wild-type p53 in HepG2 cells (Fig. 5d and Extended Data Fig. 6b)”, Extended Data Fig. 6b should be “Extended Data Fig. 6c”. In addition, IgG and p53 antibody should be noted in Extended Data Fig. 6c.

Response: Thanks for the kind reminding. We had corrected the mistakes.

(10) In fig 6i, why did aramchol and donafenib reduce p53 level?

Response: We had found that although aramchol combined with donafenib had achieved the most therapeutic effects in HepG2-derived xenograft tumors, there were still remaining tumor tissues in this group (Fig. 7a). These remaining tissues might harbor the characteristic of resistance to the combination therapy, and p53 reduction at this stage might be involved in the acquired resistance. Supporting this notion, our *in vitro* experiment had showed that although SCD1 level in HepG2 cell was decreased during the short-time exposure to sorafenib, together with slightly increase of p53 proteins, longer sorafenib treatment could lead to p53 reduction and SCD1 recovery (Additional Fig. 4). These acquired-resistance might due to heterogenous status of tumor cells before sorafenib treatment or the contribution of the tumor microenvironment *in vivo*, which needed further experiment to explore.

Additional Figure 4. The response of sorafenib treatment in HepG2 cells. The cells were treated with sorafenib for the indicated times, the remaining alive cells were collected and the proteins were measured.

(11) In fig 7a, e and Extended Data Fig. 10a, the intensities of the fluorescence signals of certain panels should be enhanced. It is hard to recognize the signals.

Response: Thanks for the kind reminding. We had enhanced the fluorescence signals of the panels. The related panels in the revised version were Fig. 8a and Extended Data Fig. 11a.

(12) Delete the full stop “.” in the end of the title and several captions in the results part.

Response: We had deleted the full stop “.” in the end of the title and the captions in the results part. We are sorry for this mistake.

(13) In this sentence “URI depletion significantly increased the sensitivity of JHH1 and HepG2 cells to sorafenib (Extended Data Fig. 2c), with a decreased in IC50”, “decreased” should be “decrease”. Or you can delete the word “in”.

Response: We had deleted the word “in” in this sentence. We thank for the reviewer’s kind reminding.

(14) In this sentence “Lipid metabolic reprogram is involved in cancer drug resistance”, “reprogram” should be “reprogramming”.

Response: We had changed “reprogram” into “reprogramming” and we thank for the kind reminding by the reviewer.

(15) In this sentence “we found that URI high-expression in HCC is associated with cancer malignant and poor survival of patients”, “malignant” should be “malignancy”.

Response: We had changed “malignant” into “malignancy” in this sentence. We thank for the reviewer’s kind reminding.

(16) In this sentence “This may helpful to keep p53 levels low as has been detected in cancer cells”, “may” should be “may be”.

Response: We had modified this sentence and we are sorry for this mistake.

(17) In this sentence “URI-p53-SCD1 axis mediates resistance of TKIs and may explain why p53-wild type HCC still showed intrinsic resistant to TKIs”, “resistant” should be “resistance”.

Response: We had changed this mistake in the sentence.

Thank you sincerely for your guidance on our article. We apologize again for the mistakes.

Reviewer #3 - URI - (Remarks to the Author):

This is an interesting manuscript suggesting that sorafenib and other TKIs may be more effective in HCC when cells are sensitized by SCD1 inhibitors, the combination of which causes ferroptosis. One strength of the manuscript is that it spans the gamut of experiments done in cell culture, xenografts and human patient samples. The link between levels of URI and SCD1 are convincing.

Specific comments:

There are typos throughout the manuscript. Please fix all of these.

Response: Thanks for your careful review. We are very sorry for our incorrect writing and the typos has been corrected in the revised manuscript.

Results section 1

The full list of genes altered in control versus shURI HepG2 cells should be provided in a table.

Response: Thanks for your kind suggestion. The differential genes between control and shURI group of HepG2 cells have been listed in Supplementary Table 1 in the revised manuscript.

Please state how many genes are related to ferroptosis and what percentage of these are affected by URI KD.

Response: Thanks for your constructive suggestion. A total of 40 genes associated with ferroptosis were detected in shURI versus control HepG2 cells (Extended Data Fig. 1c), of which 15% (5 genes for up-regulated and 1 gene for down regulated) changed significantly after URI depletion.

Please reference and describe the SCD1 promoter used in the luciferase construct.

Response: Thanks for your constructive suggestions. The 1547-bp sequence above the TSS site of SCD1 was inserted in a pGL3-Basic vector using the specific enzymes at *KpnI* and *XhoI* sites.

Extended data 5, there are two panels labeled “b”, but no panel “c”.

Response: Thanks for your careful review of our manuscript. These two panels have been corrected in the revised manuscript.

Figure 5. There is no panel labeled “n”.

Response: Thank you very much for your suggestion. We had corrected this mistake in the revised manuscript.

Figure 6h. The p53 IHC is not possible to interpret and should be improved.

Response: Thank you for your helpful suggestion. We have optimized the IHC experimental conditions so that the expression of p53 can be interpreted clearly (Fig. 7d).

Figure 6e. There is a large effect of Donafenib alone on these tumors. In light of this fact, please explain/justify how this could be a good model to study synergy between Donafenib and Aramchol.

Response: Thank you for your thoughtful advice. We had tested the therapeutic effect in the later-stage of xenograft model when the tumors reached about 200 mm³. As demonstrated in the Fig. 7, donafenib alone displayed good antitumor effects in the early tumor stage. However, in the advanced stage of tumor progression, donafenib single treatment showed limited effects of antitumor activity. In contrast, donafenib in combination with aramchol showed effective anti-tumor ability in advanced stage tumors. Considering TKI therapy is usually used for patients with advanced diseases in clinical application, this combination therapy may represent a promising strategy for the patients with advanced HCC who have wild-type p53 and high levels of URI/SCD1.

Figure 7a It is very difficult to see the IHC. Please make brighter.

Response: We appreciate this suggestion. We have made the IHC images brighter in the revised manuscript.

Figure 7 d,h,f and g. The color scheme shown for the interpretation of the graphs are different than the colors on the graphs and it is difficult to interpret the data because of this.

Text- Please define para-tumor.

Response: Thank you very much for your careful reading. The colors of the graphs have been updated in order to distinguish data of different groups (Fig. 8d, Extended Data Fig. 11c-e).

The definition of “para-tumor” as the adjacent normal-like tissue has been added in Figure Legend 8a.

Given the hypothesis that URI interacts with TRIM28 to promote p53 ubiquitination and degradation involving MDM2, it would be important to determine how the levels of URI and SCD1 correlate with levels of p53 in patient samples shown in Figure 7.

Response: Thanks for your helpful comment and we agree with your opinion. To address this question, we first employed a new HCC cohort enrolled by Gao et.al (Cell, PMID: 31585088), which we named “Fudan_HCC_cohort”. The results showed that *SCD1* levels were higher in *URI*^{high} HCC patients with p53-WT (Figure 8e). In contrast, the expression of *URI* was not associated with the expression of *SCD1* in p53-mutant HCC patients (Extended Data Figure 10f). Meanwhile, p53-WT HCC patients with high expression of URI or SCD1 had a worse prognosis (Figure 8f, g), but no such prognostic difference was observed in p53-mutant HCC patients (Extended Data Fig 10g, h). These results suggest the critical role of URI-SCD1 in p53-WT HCC progression.

We next added a new cohort (Cohort C) containing patients with recurrent HCC treated with sorafenib to explore the correlation between URI and SCD1 expression and p53 expression using formalin-fixed paraffin-embedded (FFPE) tissue sections. According to the targeted DNA sequencing data, we divided the patients into p53-WT and p53-mutant groups. In p53-WT group, there was a significant negative correlation between p53 and SCD1 expression and a positive correlation between URI and SCD1 expression (Figure 8j, k). In addition, the expression of URI and SCD1 were also linked to clinical outcome in patients receiving sorafenib treatment in the p53-WT group. We did not find correlation between URI and SCD1 expression in the p53-mutation group. These results suggested that URI-SCD1 axis was involved in sorafenib resistance in p53-WT HCC.

Reviewers' Comments:

Reviewer #1:

Remarks to the Author:

The authors have carefully addressed all the suggested points, complementing and strengthening their data with new experiments and analysis. The inclusion of the human p53 wild type and mutant HCC organoids has been properly validated and reinforces their previous findings. It also helped to effectively address other reviewers' points. The quality of the figures has notably improved, and the typographic mistakes have been corrected. The paper should be accepted for publication as the findings could be significant in the translational field.

Reviewer #2:

Remarks to the Author:

All the issues have been well addressed.

Reviewer #3:

Remarks to the Author:

The work is now acceptable for publication following this major revision.

Review report

REVIEWER COMMENTS

Reviewer #1 (Remarks to the Author):

The authors have carefully addressed all the suggested points, complementing and strengthening their data with new experiments and analysis. The inclusion of the human p53 wild type and mutant HCC organoids has been properly validated and reinforces their previous findings. It also helped to effectively address other reviewers' points. The quality of the figures has notably improved, and the typographic mistakes have been corrected. The paper should be accepted for publication as the findings could be significant in the translational field.

Response: Thank you very much for your comments and advices. These advices will also be of great help to our clinical research work in the future.

Reviewer #2 (Remarks to the Author):

All the issues have been well addressed.

Response: Thank you very much for all your comments. These comments and advices are critical to our research.

Reviewer #3 (Remarks to the Author):

The work is now acceptable for publication following this major revision.

Response: Thank you very much for your critiques, time, efforts and patience in this process. Your constructive suggestions are critical to our research.